# MULTIBAND: MULTI-TASK SONG GENERATION WITH PERSONALIZED PROMPT-BASED CONTROL

## ABSTRACT

Song generation focuses on producing controllable high-quality songs based on various personalized prompts. However, existing methods struggle to generate high-quality vocals and accompaniments with effective style control and proper alignment. Additionally, they fall short in supporting various personalized tasks based on diverse prompts. To address these challenges, we introduce MultiBand, the first multi-task song generation model for synthesizing high-quality, aligned songs with extensive control based on diverse personalized prompts. MultiBand comprises these primary models: 1) VocalBand, a decoupled model, leverages the flow-matching method for singing styles, pitches, and mel-spectrograms generation, allowing fast and high-quality vocal generation with high-level control. 2) Accomp-Band, a flow-based transformer model, incorporates the Aligned Vocal Encoder, using contrastive learning for alignment, and Band-MOE, selecting suitable experts for enhanced quality and control. This model allows for generating controllable, high-quality accompaniments perfectly aligned with vocals. 3) Two generation models, LyricBand for lyrics and MelodyBand for melodies, contribute to the comprehensive multi-task song generation system, allowing for extensive control based on multiple personalized prompts. Experimental results demonstrate that Multi-Band performs better over baseline models across multiple tasks using objective and subjective metrics. Audio samples are available at https://multiband.github.io.

## 1 INTRODUCTION

Song generation focuses on producing complete musical pieces based on text prompts (about lyrics, melodies, singing styles, and music styles), along with optional audio prompts. Unlike singing voice synthesis (SVS) (Shi et al., 2022; Cho et al., 2022; Zhang et al., 2024), which focuses on the singing component, or music generation (Dong et al., 2018; Agostinelli et al., 2023; Huang et al., 2023; Copet et al., 2024) for only instrumental tracks, song generation involves synthesizing both high-quality vocals and accompaniments with effective style control and proper alignment (Li et al., 2024a).

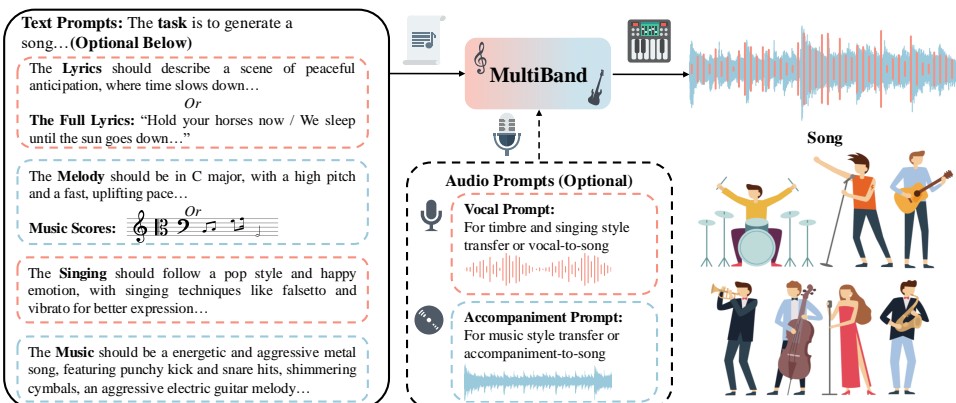

Figure 1: Overview of MultiBand, which generates complete songs like a versatile band. The dashed lines indicate optional inputs. At a minimum, users can just input "The task is to generate a song."

Despite significant advancements in SVS and music domains, generating high-quality, controllable, aligned songs remains challenging. Song generation aims to enable controllable musical experiences, with broad applications ranging from entertainment videos to professional composition. As shown in Figure 1, song generation models can leverage different prompts for multiple tasks. **Text prompts** allow for control over tasks, lyrics, melody, singing styles (like singing methods, emotion, and techniques), and music styles (like genre, tone, and instrumentation), while **audio prompts** enable users to input their voice or preferred music for customization. However, the few existing song generation models (Zhiqing et al., 2024), which typically generate vocals and accompaniments using transformer models separately, struggle to produce high-fidelity songs. These models also lack mechanisms to align vocals with accompaniments properly and fail to achieve effective control.

Currently, song generation encounters three major challenges:

- **Limitations in synthesizing high-quality vocals with effective style control.** For singing style control, StyleSinger (Zhang et al., 2024) conducts singing style transfer, while PromptSinger (Wang et al., 2024) achieves singer identity control. However, existing models have yet to generate pleasing vocals with high-level style control (like singing methods, emotion, and techniques) by text prompts, and customization with audio prompts.

- **Difficulties in generating controllable, high-fidelity, aligned accompaniments.** Existing music generation models (Copet et al., 2024) are limited to low-fidelity outputs with little control. Current text-to-song models (Zhiqing et al., 2024) also lack mechanisms for effectively aligning vocals with accompaniments. Generating controllable (like genre, tone, and instrumentation), aligned, high-quality accompaniments remains challenging.

- **Challenges in multi-task song generation based on various personalized prompts.** The limited existing song generation methods (Li et al., 2024a) primarily focus on the text-to-song task and do not support a wide range of personalized song generation tasks based on diverse text and audio prompts. This reliance on constrained inputs leads to a suboptimal user experience and restricts the models' ability to customize songs for individual preferences.

To address these challenges, we introduce MultiBand, the first multi-task song generation model for synthesizing high-quality, aligned songs with extensive control based on diverse personalized prompts. Following the human perception that accompaniment complements vocal melody with complex harmonic and rhythmic structure (Zhiqing et al., 2024), we generate them separately. To achieve fast and high-quality vocal generation with high-level control, we design a decoupled model VocalBand, predicting singing styles, pitches, and mel-spectrograms based on the flow-matching model. Based on the complex nature of music, we introduce a flow-based transformer model AccompBand to generate high-fidelity, controllable, aligned accompaniments. For proper alignment, we propose the Aligned Vocal Encoder using contrastive learning, encoding vocals to carry style, rhythm, and melody closely related to accompaniment. We also design Band-MOE (Mixture of Experts), selecting suitable experts for enhanced quality and control, considering the noise level, text prompts, and vocal embedding. Additionally, we add two generation models, LyricBand for lyrics and MelodyBand for melodies, contributing to the comprehensive multi-task song generation system.

Our experiments on a combination of open-source and web-crawled bilingual song datasets show MultiBand can generate high-quality songs based on various prompts, outperforming baseline models in multiple tasks, including melody and lyric generation, vocal generation, accompaniment and song generation, and other related tasks. The main contributions of our work are summarized as follows:

- We introduce MultiBand, the first multi-task song generation model for synthesizing high-quality, aligned songs with extensive control based on diverse personalized prompts.

- We design a decoupled model VocalBand, which leverages the flow-matching model to generate singing styles, pitches, and mel-spectrograms, enabling fast and high-quality vocal synthesis with high-level personalized control through input text and audio prompts.

- We propose a flow-based transformer model AccompBand to generate high-quality, controllable, aligned accompaniments, with the Aligned Vocal Encoder, using contrastive learning for alignment, and Band-MOE, selecting suitable experts for enhanced quality and control.

- Experimental results demonstrate that MultiBand enables high-level personalized control across multiple bilingual song generation tasks based on various text and audio prompts, achieving superior objective and subjective evaluations compared to baseline models.

## 2 BACKGROUND

**Singing Voice Synthesis**  Singing Voice Synthesis (SVS) rapidly advances as a field for generating high-quality singing voices from given lyrics and music scores. Choi & Nam (2022) presents a melody-unsupervised model that only requires pairs of audio and lyrics, eliminating the need for temporal alignment. Wesinger (Zhang et al., 2022c) proposes a Transformer-alike acoustic model. VISinger 2 (Zhang et al., 2022b) employs digital signal processing techniques to enhance synthesis fidelity, while Kim et al. (2024) uses adversarial multi-task learning to disentangle timbre and pitch, improving the naturalness of generated voices. Further advancements include MuSE-SVS (Kim et al., 2023), which introduces a multi-singer emotional singing voice synthesizer, and StyleSinger (Zhang et al., 2024), which facilitates style transfer and zero-shot synthesis by using audio prompts to extract timbre and styles via a residual quantization method. Additionally, PromptSinger (Wang et al., 2024) attempts to control speaker identity in singing voices based on text descriptions. On the dataset front, M4Singer (Zhang et al., 2022a) and OpenSinger (Huang et al., 2021) contribute by releasing multi-singer datasets. Despite these advancements, these approaches can not generate accompaniment aligned with vocals. Recently, Melodist (Zhiqing et al., 2024) has introduced a text-to-song model that sequentially generates vocal and accompaniment codec tokens using two auto-regressive transformers. However, the challenge of achieving highly controllable and personalized pleasing vocals persists.

**Music Generation**  Music generation encompasses multiple tasks, like symbolic music generation and accompaniment creation, based on text descriptions or audio prompts. MuseGAN (Dong et al., 2018) employs a GAN-based approach to generate symbolic music, while PopMAG (Ren et al., 2020a) generates many instrumental tracks simultaneously. SongMASS (Sheng et al., 2021) uses transformer models to generate lyrics or melodies conditioned on each other. MusicLM (Agostinelli et al., 2023) leverages joint textual-music representations from MuLan (Huang et al., 2022a) to generate semantic and acoustic tokens based on transformer decoders. MusicGen (Copet et al., 2024) generates music codec tokens within a single transformer decoder with codebook interleaving patterns. MusicLDM (Chen et al., 2024) incorporates beat-tracking information and employs data augmentation through latent mixups to address potential plagiarism concerns in music generation. Additionally, SingSong (Donahue et al., 2023) presents a model capable of generating background music to complement provided vocals. Recently, MelodyLM (Li et al., 2024a) has employed transformer models for melody and vocal generation, along with a latent diffusion model to create accompaniments. Nevertheless, challenges remain in improving the controllability and quality of music generation. Existing methods also lack the necessary mechanisms for precisely aligning vocals and accompaniments, and they do not support multi-task song generation based on various personalized text and audio prompts.

## 3 METHOD

This section introduces MultiBand. We design two distinct models, VocalBand for vocals and AccompBand for accompaniments, tailored to their unique characteristics. Additionally, we incorporate LyricBand for lyrics and MelodyBand for melodies, composing a multi-task song generation system.

### 3.1 MULTI-TASK SONG GENERATION

As shown in Figure 2 (a), MultiBand handles multi-task song generation based on text and audio prompts. We employ a text encoder to generate text tokens $z_p$. Without lyrics or music scores, LyricBand and MelodyBand predict phonemes $p$ and notes $n$ (pitch and duration) as target contents. Next, in Figure 2 (b), to achieve fast and high-quality vocal generation with granular and personalized control, we introduce VocalBand, which decouples the content $z_c$, timbre $z_t$, and style $z_s$. Through the Flow-based Pitch Predictor, Mel Decoder, and pre-trained vocoder, the target vocal $y_v$ is synthesized. Then, in Figure 2 (c), for the complex nature of accompaniment, we design AccompBand to achieve superior quality, control, and alignment. AccompBand uses Aligned Vocal and Accomp Encoders, pre-trained through contrastive learning, to extract aligned embeddings $z_v$ from $y_v$ and $x$ from ground truth (GT) accompaniment $\hat{y}_a$ during training. $z_v$ and noise-injected $x_t$ are processed by multiple Band Transformer Blocks with Band-MOE (Mixture of Experts), which selects suitable experts based on $z_v$, $z_p$, and time step $t$ for enhanced quality and control. During inference, the ordinary differential equation (ODE) solver, Aligned Accomp Decoder, and vocoder generate the target accompaniment $y_a$ from input $z_v$ and Gaussian noise $\epsilon$. Finally, $y_v$ and $y_a$ are combined to the final target song $y$.

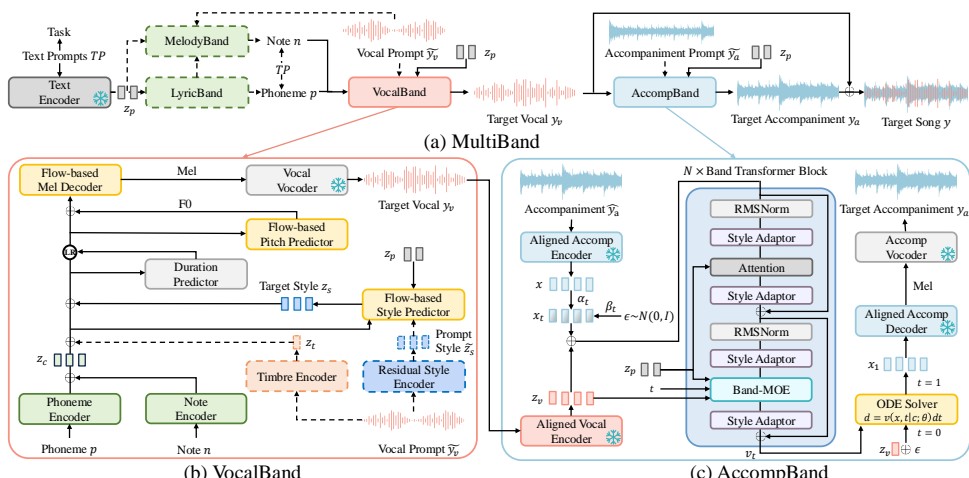

Figure 2: The overall architecture of MultiBand. Vocal and accompaniment are generated separately by the VocalBand and AccompBand. Dashed lines represent optional processes, while LR stands for length regulator. Modules marked with a snowflake icon are frozen during the training phase.

## 3.2 VOCALBAND

**Decomposition** As shown in Figure 2(b), for more personalized and fine-grained control, we disentangle target vocal $y_v$ into distinct representations: content $z_c$, style $z_s$ (e.g., singing methods, emotion, techniques, pronunciation, articulation skills), and timbre $z_t$. For $z_c$, phonemes $p$ and notes $n$ (note pitch and duration) are encoded by a phoneme encoder and a note encoder. Given a vocal prompt $\tilde{y}_v$, the timbre and personalized styles (like pronunciation and articulation skills) should remain consistent. We pass $\tilde{y}_v$ through a timbre encoder to obtain $\tilde{z}_t$, while $z_t = \tilde{z}_t$. Next, the residual style encoder employs a Residual Quantization model (Lee et al., 2022a) to extract phoneme-level prompt style $\tilde{z}_s$. This serves as an information bottleneck to filter out non-style information (Zhang et al., 2024), ensuring effective decomposition. The Flow-based Style Predictor uses $z_c$, $z_t$, $\tilde{z}_s$, and text tokens $z_p$ to predict $z_s$, learning both personalized styles of $\tilde{z}_s$ and style control information in $z_p$ (like singing methods, emotions, techniques). For more details, please refer to Appendix C.2.

**Flow-based Style Predictor** Singing styles typically exhibit continuous and complex dynamics, involving intricate variations. The flow-matching model (Liu et al., 2022b) is suitable for generating styles with finer-grained control by modeling styles as a smooth transformation, effectively balancing multiple control inputs, enabling a fast and stable generation of natural and consistent styles.

As shown in Figure 3 (a), we design the Flow-based Style Predictor using content $z_c$, timbre $z_t$, prompt style $\tilde{z}_s$, and text tokens $z_p$ to predict the target style $z_s$. With input $z_c$ and $z_t$, we employ a style alignment model with the Scaled Dot-Product Attention mechanism (Vaswani et al., 2017) to align style control information from $z_p$ (e.g., singing methods, emotions, techniques) with contents. The fused condition $c$ is then fed into an ODE solver, which transforms Gaussian noise $\epsilon$ into $z_s$ along a probability path $p_t(z_{st})$. We concatenate $\tilde{z}_s$ with $\epsilon$ to allow $z_s$ to learn personalized styles (e.g., pronunciation, articulation skills). $z_{st}$ is obtained by linear interpolation at time $t$ between $\epsilon$ and $z_s$, which is extracted from the GT vocal by the residual style encoder, thus the target vector field $u(z_{st}, t) = z_s - \epsilon$. The learned vector field $v_t(z_{st}, t|c; \theta)$, predicted by a vector field estimator at each time $t$, ensures smooth interpolation between the initial noise and output $z_s$, guided by the flow-matching objective, which minimizes the distance between the learned and true vector fields:

$$\mathcal{L}_{style} = \mathbb{E}_{t, p_t(z_{st})} \left\| v_t(z_{st}, t|c; \theta) - (z_s - \epsilon) \right\|^2. \tag{1}$$

where $p_t(z_{st})$ represents the distribution of $z_{st}$ at time $t$. This method ensures the fast and controlled generation of phoneme-level target style $z_s$, learning both personalized styles consistent with $\tilde{z}_s$ and aligned style control information from $z_p$. For more details, please refer to Appendix A and C.6.

**Flow-based Pitch Predictor and Mel Decoder** Traditional pitch predictors and mel decoders struggle to capture the dynamic and complex variations in singing voices. To overcome these, we

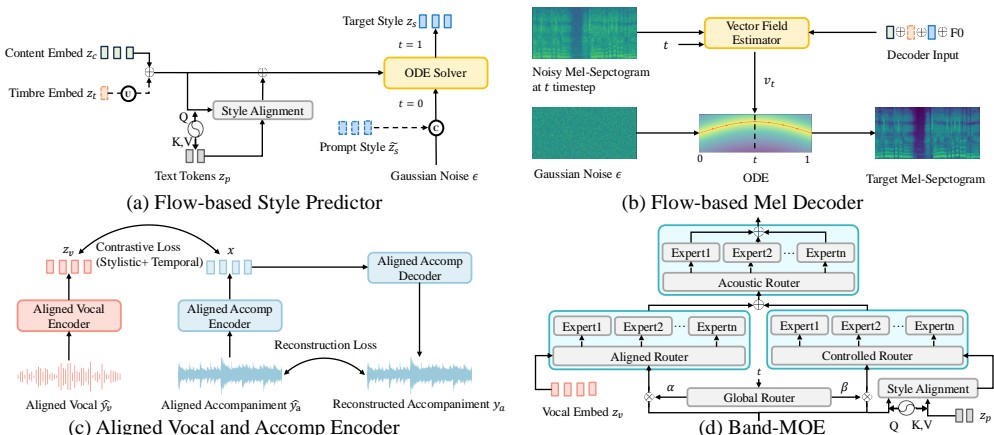

Figure 3: The architecture of four major components of VocalBand and AccompBand. Dashed lines represent optional processes. C and U represent concatenation and upsampling operations.

propose the Flow-based Pitch Predictor and Mel Decoder, which use content $z_c$, timbre $z_t$, and style $z_s$ to quickly and robustly generate high-quality F0 and mel-spectrograms. As shown in Figure 3(b), the Flow-based Mel Decoder employs a flow-matching architecture (Liu et al., 2022b), where the vector field estimator and ODE solver generate the target mel-spectrogram from Gaussian noise $\epsilon$. The Flow-based Pitch Predictor follows a similar flow-matching procedure. Our pitch loss $\mathcal{L}_{pitch}$ and mel loss $\mathcal{L}_{mel}$ are analogous to $\mathcal{L}_{style}$ in Equation 1. For more details, please refer to Appendix C.7.

### 3.3 ACCOMPBAND

**Aligned Vocal and Accomp Encoders**  The style, rhythm, and melody of a song are complex and variable, making the alignment between vocal and accompaniment challenging in the song generation task. As shown in Figure 3 (c), given a vocal-accompaniment pair $(\hat{y}_v, \hat{y}_a)$, we design the Aligned Vocal and Accomp Encoders to extract aligned embeddings $(z_v, x)$. To reconstruct high-quality accompaniment, we introduce the Aligned Accomp Decoder. All encoders and decoders are based on the VAE model (Kingma & Welling, 2013). For better alignment, we employ the contrastive objective (Radford et al., 2021) and define two types: stylistic contrast $\mathcal{L}_{sty}$ and temporal contrast $\mathcal{L}_{tem}$. For $\mathcal{L}_{sty}$, we maximize the similarity of pairs from the same song and minimize for different songs. It encourages learning stylistic alignment across different songs. In specific, we encode pairs from different songs, $\mathcal{B} = \{(x^i, z_v^i)\}_{i=1}^N$, where $N$ is the number of songs. We define $\mathcal{L}_{sty}$ as:

$$\mathcal{L}_{sty} = -\frac{1}{2N} \sum_{i=1}^N \left( \log \frac{\exp(sim(x^i, z_v^i)/\tau)}{\sum_{j=1}^N \exp(sim(x^i, z_v^j)/\tau)} + \log \frac{\exp(sim(x^i, z_v^i)/\tau)}{\sum_{j=1}^N \exp(sim(x^j, z_v^i)/\tau)} \right), \quad (2)$$

where $sim(\cdot)$ denotes cosine similarity. For $\mathcal{L}_{tem}$, we encode pairs from different time segments within the same song. The objective is to maximize the similarity of pairs from the same time segment and minimize for different time segments. We define a temporal contrast objective $\mathcal{L}_{tem}$, similar to the stylistic contrast $\mathcal{L}_{sty}$ in Equation 2. For training the Aligned Accompaniment Decoder, we use L2 reconstruction loss $\mathcal{L}_{recon}$ and employ a GAN discriminator with LSGAN-styled adversarial loss $\mathcal{L}_{adv}$ (Mao et al., 2017) for better reconstruction. After training, we can encode an audio-video pair $(\hat{y}_v, \hat{y}_a)$ into a highly aligned embedding pair $(z_v, x)$, with the $z_v$ containing style, rhythm, and melody closely related to accompaniment. We pre-train these encoders and decoders before training AccompBand, facilitating subsequent generation. For more details, please refer to Appendix D.2.

**Band-MOE**  Accompaniment generation is highly complex due to the intricate interplay of various instruments and alignment with vocals, especially for high-quality and long-sequence generation. Flow matching enables smooth transformations, leading to stable and quick generation, while transformer models effectively capture intricate long-range dependencies, making flow-based transformers suitable for this task. We integrate the Aligned Vocal Encoder's output $z_v$ with the noisy input $x_t$, utilizing the self-attention mechanism of the transformer for alignment. Based on Flag-Dit (Gao et al., 2024), we design and stack multiple Band Transformer Blocks as the vector field estimator.

For enhanced music quality and control, we propose Band-MOE (Mixture of Experts). As shown in Figure 3 (d), Band-MOE consists of three expert groups: Aligned MOE, Controlled MOE, and Acoustic MOE, each containing multiple experts. Aligned MOE conditions on $z_v$, adjusting inputs to match vocal features like loudness and frequency, selecting suitable experts like one specialized in large loudness and alto range. Controlled MOE uses aligned styles in text prompts to select experts for fine-grained style control, such as one for aggressive drums with metal guitar tones. Given the varying behavior of the transformer at different noise levels (Feng et al., 2023), we design a global router that adjusts the weighting of outputs from Aligned MOE and Controlled MOE: 1) at early time steps (near 0), where hidden representation $h$ is highly noisy, the network prioritizes matching with vocal for coherent reconstruction; 2) at later time steps (near 1), where $h$ has been largely reconstructed, the network focuses more on refining stylistic details, relying heavily on text prompts.

Finally, mel-spectrogram patterns exhibit variation across acoustic frequencies (Lee et al., 2022b). In music, high-frequency components often include the harmonics and overtones of instruments like strings and flutes. At the same time, low-frequency content typically encompasses basslines and kick drums providing rhythm and depth. Since the Aligned Vocal and Accomp Encoders employ 1D convolutions, the latent should retain the frequency distribution. Therefore, we design Acoustic MOE, selecting experts by different acoustic frequency dimensions for better quality. All routing strategies are based on the dense-to-sparse Gumbel-Softmax (Nie et al., 2021), allowing dynamic and efficient expert selection. For more model details and MOE algorithm, please refer to Appendix D.3 and D.4.

**Classifier-free Guidance**  To further control styles of the generated accompaniment based on input text prompts, we implement the classifier-free guidance (CFG) strategy. During AccompBand training, we randomly replace input text tokens $z_p$ with encoded empty strings $\varnothing$ at a probability of 0.2. During inference, we modify the output vector field of the Band Transformer blocks as follows:

$$v_{cfg}(x,t|z_p;\theta) = \gamma v_t(x,t|z_p;\theta) + (1-\gamma)v_t(x,t|\varnothing;\theta), \tag{3}$$

where $\gamma$ is the classifier free guidance scale trading off creativity and controllability. When $\gamma = 1$, $v_{cfg}$ is the same as the original $v_t(x,t|z_p;\theta)$. For more details, please refer to Appendix D.5.

### 3.4 Lyric and Melody Generation

**LyricBand**  To enable more personalized song generation tasks, we introduce LyricBand, a system designed to generate complete song lyrics based on more customizable text prompts. Users can design the theme, emotion, and other parameters to generate fully personalized lyrics. We leverage QLoRA (Dettmers et al., 2024) for efficient fine-tuning of a well-performing open-source bilingual large language model Qwen-7B (Bai et al., 2023). By utilizing 4-bit quantization and low-rank adapters, QLoRA enables LyricBand to adapt effectively to lyric generation, enabling high-level customization and creativity across various input text prompts. For more details, please refer to Appendix E.1.

**MelodyBand**  Previous singing voice and song generation models often require users to provide music scores to achieve stable melodies (Zhiqing et al., 2024), lacking personalized customization of the melody. Inspired by symbolic music models (Dong et al., 2018), we propose MelodyBand, which generates musical notes based on text prompts, lyrics, and vocal prompts. We employ a non-autoregressive transformer model to efficiently generate note pitches and durations simultaneously. After encoding phonemes and timbre, MelodyBand achieves fine-grained melody control by injecting text tokens through cross-attention mechanisms. We train MelodyBand with the cross-entropy loss for note pitches and an L2 loss for note durations. For more details, please refer to Appendix E.2.

### 3.5 Training and Inference

The VocalBand, AccompBand, LyricBand, and MelodyBand are trained separately, and the detailed training details are provided in Appendix B.2. For inference, our model can accept various prompts for multiple tasks. Without full lyrics or music scores as input, LyricBand and MelodyBand generate phonemes $p$ and notes $n$ as target contents. For song generation or singing style transfer tasks, VocalBand generates the target vocal $y_v$, as well as AccompBand generates the target accompaniment $y_a$ from $y_v$ and Gaussian noise $\epsilon$. During music style transfer, AccompBand uses noisy prompt accompaniment $\tilde{y_a}$ instead of $\epsilon$ as input. In the vocal-to-song task, VocalBand is not used, whereas in the accompaniment-to-song task, notes $n$ are extracted from ground-truth accompaniment $\hat{y_a}$ to guide VocalBand for vocal generation. More inference details can be found in Appendix B.3.

## 4 EXPERIMENTS

### 4.1 EXPERIMENTAL SETUP

**Dataset**   We train our model using a combination of bilingual web-crawled and open-source song datasets. Since there are no publicly available annotated song datasets including vocals and accompaniments, we collect 20k Chinese and English songs from well-known music websites. To expand data, we also incorporate open-source singing datasets including OpenCpop (Wang et al., 2022) (5 hours in Chinese), M4Singer (Zhang et al., 2022a) (30 hours in Chinese), OpenSinger (Huang et al., 2021) (83 hours in Chinese), and PopBuTFy (Liu et al., 2022a) (10 hours in English). After processing and cleaning, we have about 1,000 hours of song data and 1,100 hours of vocal data. We also use a filtered subset of LP-MusicCaps-MSD (Doh et al., 2023), resulting in about 1,200 hours of accompaniment data. For zero-shot evaluation, we leave out 500 out-of-domain bilingual samples with unseen singers as the test set for each task. For more details, please refer to Appendix F.

**Implementation Details**   We derive mel-spectrograms from raw waveforms with a 48kHz sample rate, 1024 window size, 320 hop size, and 80 mel bins. We use 4 layers of Band Transformer Blocks. The flow-matching time step is 100 for VocalBand and 1000 for AccompBand during training, while 25 during inference with the Euler ODE solver. For more details, please refer to Appendix B.1.

**Evaluation Metrics**   We conduct both subjective and objective evaluations on generated samples. For lyric generation, we use overall quality (OVL) and relevance to the prompt (REL) for subjective evaluation. In melody generation, multiple objective metrics are employed for testing controllability. We use the Krumhansl-Schmuckler algorithm to predict the potential key of the generated notes and report the average key accuracy KA. We compute the average absolute difference of average pitches (APD) and temporal duration (TD, in seconds). Moreover, we employ pitch and duration distribution similarity (PD and DD). Melody distance (MD) is also computed using dynamic time warping.

For vocal generation, we conduct MOS (Mean Opinion Score) as the subjective evaluation. We use MOS-Q for synthesized quality and MOS-C for controllability based on text prompts. We also use F0 Frame Error (FFE) as the objective metric. For singing style transfer, we also employ MOS-S and Cosine Similarity (Cos) to assess singer similarity in timbre and personalized styles of vocal prompts.

For accompaniment and song generation, we ask raters to evaluate the audio samples in terms of overall quality (OVL), relevance to the prompt (REL), and alignment with the vocal (ALI). For objective evaluation, we calculate Frechet Audio Distance (FAD), Kullback–Leibler Divergence (KLD), and the CLAP score (CLAP). Please refer to Appendix G for more details about evaluation.

**Baseline Models**   For lyric generation, we use the original Qwen-7B (Bai et al., 2023) as the baseline model. For melody generation, we compare with SongMASS (Sheng et al., 2021) and MIDI part of MelodyLM (Li et al., 2024a). For vocal generation, we compare against VISinger2 (Zhang et al., 2022b), a traditional high-fidelity SVS model, StyleSinger (Zhang et al., 2024), the current best zero-shot SVS model with style transfer, and vocal parts of Melodist (Zhiqing et al., 2024) and MelodyLM. For accompaniment generation, we compare with MusicGen (Copet et al., 2024), LuminaT2Music (Gao et al., 2024), and the accompaniment parts of Melodist and MelodyLM. For closed-source models Melodist and MelodyLM, we report objective metrics in their papers and use their demo pages for subjective evaluation. For other models, we employ their open-source codes.

### 4.2 LYRIC AND MELODY GENERATION

**Lyric Generation**   We aim to build a comprehensive multi-task song generation system based on more personalized text prompts. Given the absence of models specifically designed for generating bilingual lyrics based on text prompts, we fine-tune the well-performing, open-source bilingual language model Qwen-7B on lyrics of our datasets using QLoRA to enhance its lyric generation capabilities. We experiment with different text prompts covering aspects such as theme, emotion, genre,

Table 1: Results of lyric generation.

| Methods | OVL↑ | REL↑ |
|---|---|---|
| GT | 92.31±1.29 | 84.07±1.63 |
| Qwen-7B | 74.35±1.37 | 80.66±0.92 |
| LyricBand | **79.68±1.05** | **82.01±1.13** |

Table 2: Results of melody generation.

| Methods | KA(%)↑ | APD↓ | TD↓ | PD(%)↑ | DD(%)↑ | MD↓ |
|---|---|---|---|---|---|---|
| SongMASS | 58.9 | 3.78 | 2.93 | 55.4 | 68.1 | 3.41 |
| MelodyLM (w/o prompt) | 54.3 | 3.61 | 5.41 | 53.9 | 26.4 | 4.32 |
| MelodyLM | **76.6** | 2.05 | 2.29 | 62.8 | 40.8 | 3.62 |
| MelodyBand (w/o all prompts) | 50.8 | 2.06 | 2.49 | 47.1 | 60.2 | 3.39 |
| MelodyBand (w/o text prompt) | 53.1 | 1.98 | 1.73 | 51.1 | 60.3 | 3.25 |
| MelodyBand (w/o audio prompt) | 64.6 | 1.90 | 1.73 | 65.4 | 67.7 | 3.34 |
| MelodyBand | 72.7 | **1.74** | **1.65** | **65.8** | **70.5** | **3.12** |

Table 3: Results of vocal generation and singing style transfer.

| Methods | Vocal Generation | | | Singing Style Transfer | | | |
|---|---|---|---|---|---|---|---|
| | MOS-Q ↑ | MOS-C ↑ | FFE ↓ | MOS-Q ↑ | MOS-C ↑ | MOS-S ↑ | Cos ↑ |
| GT | 4.34 ± 0.09 | - | - | 4.35 ± 0.06 | - | - | - |
| Melodist | 3.83±0.09 | - | 0.12 | - | - | - | - |
| MelodyLM | 3.88±0.10 | - | 0.08 | 3.76±0.12 | - | 3.81±0.12 | - |
| VISinger2 | 3.62±0.07 | 3.63±0.09 | 0.16 | 3.55±0.11 | 3.57±0.05 | 3.70±0.08 | 0.82 |
| StyleSinger | 3.90±0.08 | 3.96±0.05 | 0.08 | 3.87±0.06 | 3.86±0.09 | 4.05±0.05 | 0.89 |
| VocalBand | **4.04±0.08** | **4.02±0.07** | **0.07** | **3.96±0.10** | **3.95±0.06** | **4.12±0.04** | **0.90** |

style, and specific keywords to generate lyrics. Then, we conduct subjective evaluations for the quality of lyrics and relevance to prompts. As shown in Table 1, our fine-tuned LyricBand model outperforms the original Qwen-7B model in overall quality and relevance to text prompts. This highlights the effectiveness of our LyricBand in handling specific downstream tasks more proficiently.

**Melody Generation**    For MelodyLM, since the melody part is closed-sourced, we directly use the objective metrics reported in the paper. Meanwhile, following MelodyLM, we add versions of MelodyBand without text or audio prompts as baseline models. As shown in Table 2, MelodyBand outperforms SongMASS across all metrics and performs better than MelodyLM in metrics except KA. The inclusion of text and audio prompts significantly improves the controllability, while removing the prompts allows for more creative freedom. Since we use a non-autoregressive transformer architecture, the generation speed is much faster than the autoregressive generation of the multi-scale transformer architecture used by MelodyLM. Therefore, although MelodyLM has a slightly higher KA, our architecture is more suitable for our comprehensive multi-task song generation system.

## 4.3   VOCAL GENERATION

**Vocal Generation**    The same test set with unseen singers is used for VISinger2 and StyleSinger in zero-shot vocal generation. Additionally, to enable style control (like singing method, emotion, and techniques), we incorporate our text encoder and style alignment models into these systems to capture style control information aligned with contents. We provide timbre embedding (Wan et al., 2018) to these models; for VocalBand, we leverage vocal prompts only providing timbre for comparison. For the vocal parts of Melodist and MelodyLM, we report the objective metrics in their papers and use their demos for subjective evaluation. Notably, Melodist uses known singer IDs, making it unfair for the zero-shot comparison. Additionally, neither Melodist nor MelodyLM control singing styles, therefore MOS-C is not provided. As shown in Table 3, VocalBand outperforms other models in both quality (MOS-Q, FFE) and controllability (MOS-C). This demonstrates the effectiveness of the Flow-based Style Predictor for style control and the high quality provided by the Flow-based Pitch Predictor and Mel Decoder. For more detailed and visualized results, please refer to Appendix H.1.

**Singing Style Transfer**    We use vocal segments different from the target but by the same unseen singer in the test set as vocal prompts, providing timbre and personalized styles (e.g., pronunciation and articulation skills) to transfer. Baseline models are configured in the same way as in the vocal generation task, except Melodist since it does not conduct zero-shot generation. As shown in Table

Table 4: Results of accompaniment generation.

| Methods | FAD ↓ | KLD ↓ | CLAP ↑ | OVL ↑ | REL ↑ | ALI ↑ |
|---|---|---|---|---|---|---|
| MusicGen | 3.91 | 1.38 | 0.31 | 82.26±0.92 | 84.32±1.86 | - |
| LuminaT2Music | 3.31 | 1.34 | 0.35 | 85.98±1.13 | 86.47±1.38 | - |
| Melodist | 3.80 | 1.34 | 0.39 | 84.64±0.71 | 85.97±1.51 | 74.86±1.13 |
| MelodyLM | 3.42 | 1.35 | 0.35 | 85.73±1.82 | 86.44±0.90 | 75.41±1.34 |
| AccompBand | **2.92** | **1.22** | **0.56** | **88.65±1.45** | **89.31±1.13** | **80.72±1.49** |

Table 5: Results of song generation. Content includes lyrics and music scores.

| Methods | FAD ↓ | KLD ↓ | CLAP ↑ | OVL ↑ | REL ↑ | ALI ↑ |
|---|---|---|---|---|---|---|
| Melodist | 3.81 | 1.34 | 0.39 | 84.12±1.54 | 85.97±1.51 | 74.86±1.13 |
| MelodyLM | 3.42 | 1.35 | 0.35 | 85.23±1.62 | 86.44±0.90 | 75.41±1.34 |
| MultiBand (w/o lyrics) | 3.35 | 1.30 | 0.49 | 86.82±0.93 | 86.03±1.04 | 77.06±1.28 |
| MultiBand (w/o scores) | 3.39 | 1.31 | 0.46 | 85.18±0.71 | 85.04±1.25 | 75.83±1.67 |
| MultiBand (w/o contents) | 3.43 | 1.34 | 0.40 | 84.19±1.18 | 84.71±1.91 | 75.41±1.53 |
| MultiBand (w/o prompts) | 3.58 | 1.36 | - | 83.37±1.23 | - | 74.94±1.69 |
| MultiBand | **3.03** | **1.26** | **0.55** | **87.66±1.34** | **87.95±0.79** | **80.72±1.49** |

3, VocalBand outperforms baseline models in quality (MOS-Q), similarity (MOS-S and Cos), and controllability (MOS-C). This demonstrates that, when given a vocal prompt, VocalBand not only achieves style control based on text prompts but also transfers timbre and personalized styles from the vocal prompt. This highlights the excellent performance of the Flow-based Style Predictor for style control and style transfer. For more detailed and visualized results, please refer to Appendix H.2.

## 4.4 ACCOMPANIMENT AND SONG GENERATION

**Accompaniment Generation**  We use the same accompaniment dataset to train MusicGen and LuminaT2Music, and since these models do not use vocals as a condition to generate music, we only compare them in terms of quality and controllability. For the accompaniment parts of MelodyLM and Melodist, we report the objective metrics in their papers and use their demo pages for subjective evaluation. We randomly choose text prompts with various styles (e.g., genre, tone, and instrumentation) for subjective evaluation. As reported in Table 4, AccompBand outperforms baseline models in both quality (FAD, KLD, OVL) and controllability (CLAP, REL). This highlights the improvements made by Controlled MOE in style control through text prompts, resulting in a lower REL. It also demonstrates that the Acoustic MOE is effective at modeling features with complex acoustic channels of music, as reflected in the lower FAD. The highest ALI score further indicates that the Aligned Vocal and Accomp Encoders, along with Aligned MOE significantly enhance vocal alignment.

**Song Generation**  For an ultimate evaluation, we remix the generated vocals by VocalBand and accompaniments by AccompBand. For MelodyLM and Melodist, we still use the objective metrics provided in their papers and subjectively evaluate the demos available on their demo pages. We test the multi-task capabilities of AccompBand under different input conditions: incorporating LyricBand when full lyrics are not provided, adding MelodyBand when music scores are missing, using both LyricBand and MelodyBand when contents (both lyrics and music scores) are not provided, testing the case with no prompts excluding the task, and finally evaluating the scenario where all text prompts are provided. Notably, since MelodyLM and Melodist do not use text prompts to control singing styles, their REL score only considers accompaniment controllability. In contrast, for our model, we evaluate controllability for lyrics, melody, singing styles, and music styles based on text prompts.

The results are listed in Table 5, where MultiBand demonstrates the highest perceptual quality (FAD, KLD, OVL), the best adherence to text prompts (CLAP, REL), and the most effective alignment between vocals and accompaniments (ALI). This demonstrates the quality and controllability of VocalBand, as well as the quality, controllability, and excellent vocal alignment of AccompBand. When some elements in text prompts are removed, MultiBand can strike an impressive balance between creativity and stability. For experiments about more tasks, please refer to Appendix H.

Table 6: Results of ablation study on AccompBand.

| Methods | FAD ↓ | KLD ↓ | CLAP ↑ | OVL ↑ | REL ↑ | ALI ↑ |
|---|---|---|---|---|---|---|
| AccompBand | 2.92 | 1.22 | 0.56 | 88.65±1.45 | 89.31±1.13 | 80.72±1.49 |
| w/o Aligned Encoder | 3.11 | 1.25 | 0.55 | 88.12±1.36 | 89.07±1.12 | 76.33±1.09 |
| w/o Band-MOE | 3.27 | 1.33 | 0.41 | 86.03±1.27 | 87.58±0.82 | 77.53±1.48 |
| w/o Aligned MOE | 3.16 | 1.26 | 0.55 | 87.24±1.17 | 88.54±0.69 | 77.92±1.38 |
| w/o Controlled MOE | 3.08 | 1.24 | 0.43 | 88.42±1.79 | 87.96±1.48 | 79.39±1.59 |
| w/o Acoustic MOE | 3.25 | 1.31 | 0.41 | 86.43±1.52 | 87.65±1.01 | 79.14±1.22 |

## 4.5 ABLATION STUDY

**Ablation Study on VocalBand** We conduct tests on key modules of VocalBand. To compare quality, we remove the style information from the Flow-based Style Predictor, and replace the Flow-based Pitch Predictor and Mel Decoder with simpler models from FastSpeech2 (Ren et al., 2020b) for comparison. As shown in Table 7, we observe that the absence of style representation leads to a decrease in quality, as it cannot generate vocals with rich emotional and stylistic variations, nor can it achieve style control or style transfer. Additionally, our Flow-based Pitch Predictor and Mel Decoder contribute significantly to the overall quality.

Table 7: Ablation Results of VocalBand.

| Methods | MOS-Q↑ | MOS-C↑ | FEE↓ |
|---|---|---|---|
| VocalBand | 4.04±0.08 | 4.02±0.07 | 0.07 |
| w/o styles | 3.87±0.04 | - | 0.09 |
| w/o Pirch Predictor | 3.79±0.06 | 3.99±0.09 | 0.09 |
| w/o Mel Decoder | 3.68±0.08 | 3.92±0.07 | 0.13 |

**Ablation Study on AccompBand** We conduct tests on major modules of AccompBand. We replace the Aligned Vocal Encoder with a simple linear mel encoder and replace the Aligned Accomp Encoder and Decoder with a pre-trained VAE as a baseline model. Additionally, we also set the full Band-MOE and three expert groups removed as other baseline models. As shown in Table 6, removing the Aligned Encoders leads to a significant drop in vocal alignment. Moreover, removing the Band-MOE results in a decline in all metrics. For individual expert groups, we observe that the Aligned MOE affects alignment, while the Controlled MOE impacts controllability. The absence of the Acoustic MOE, which handles different acoustic channels, leads to a drop in quality.

**Ablation Study on MultiBand** We remove various components from text prompts for evaluation. As shown in Table 5, even with a minimum input of "the task is to generate a song," without other prompts, MultiBand still delivers remarkable performance. When listening to the songs generated for various tasks on our demo page, it is evident that MultiBand demonstrates strong controllability and expressiveness over styles dictated by the text prompts, along with the ability to produce intricate, skillful vocals employing multiple techniques, and complex, well-aligned accompaniments featuring harmonious instrumentation. For ablation studies on more modules, please refer to Appendix I.

## 5 CONCLUSIONS

In this paper, we present MultiBand, the first multi-task song generation model for synthesizing high-quality, aligned songs with extensive control based on diverse personalized prompts. We mainly design these models: 1) VocalBand, a decoupled model leveraging the flow-matching model for singing styles, pitches, and mel-spectrograms generation, allowing high-level control for fast and high-quality vocal generation. 2) AccompBand, a flow-based transformer model, with the Aligned Vocal Encoder, using contrastive learning for alignment, and Band-MOE, selecting suitable experts for enhanced quality and control. This model generates controllable, high-quality accompaniments perfectly aligned with vocals. 3) Two generation models, LyricBand for lyrics and MelodyBand for melodies, contribute to the comprehensive multi-task song generation system. Experimental results demonstrate that MultiBand enables high-level personalized control across multiple song generation tasks based on various prompts, achieving superior objective and subjective evaluations compared to baseline models. Due to the space limitation, we include additional discussions in the Appendix J.

## 6 ETHICS STATEMENT

Large-scale generative models always present ethical challenges. MultiBand, due to its multi-task song generation capabilities, could potentially be misused for dubbing in entertainment short videos, raising concerns about the infringement of famous singers' copyrights. Then, its ability to transfer and control multiple song styles about lyric, melody, singing, and music, lowers the requirements for high-quality, personalized, controllable song generation, posing some risks like unfair competition and potential unemployment for professionals in related music and singing occupations. To mitigate these potential risks, we will explore methods like music watermarking to protect individual privacy.

## 7 REPRODUCIBILITY STATEMENT

We have implemented several measures to ensure reproducibility: 1) We provide very detailed explanations of each module in our Appendix B, C, D, and E. We will also release the code after the paper is accepted. 2) We offer hyperparameters and experimental configurations for each model in Appendix B.1, C.1, D.1, E.1, and E.2. 3) The data processing steps and the open-source tools we used are described in detail in F. Since our datasets consist of open-source singing voices and songs collected from the internet, we will provide all web links and corresponding text prompt annotations after the paper is accepted. 4) All evaluation metrics are thoroughly described in Appendix G.

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

# A   RECTIFIED FLOW-MATCHING

In this section, we introduce the flow-matching generative method, as described by Liu et al. (2022b). In generative modeling, the true data distribution is denoted as $q(x_1)$, which can be sampled but lacks an accessible density function. Consider a probability path $p_t(x_t)$, where $x_0 \sim p_0(x)$ represents a known simple distribution (e.g., a standard Gaussian), and $x_1 \sim p_1(x)$ approximates the real data distribution. The objective of flow-matching is to model this probability path directly, expressed as an ordinary differential equation (ODE):

$$\mathrm{d}x = u(x,t)\mathrm{d}t, \quad t \in [0,1], \tag{4}$$

where $u(x,t)$ denotes the target vector field, and $t$ is the time index. If the vector field $u$ is known, realistic data can be recovered by reversing the flow. To approximate $u$, a vector field estimator $v(\cdot)$ is used, with the flow-matching objective defined as:

$$\mathcal{L}_{\mathrm{FM}}(\theta) = \mathbb{E}_{t,p_t(x)} \|v(x,t;\theta) - u(x,t)\|^2, \tag{5}$$

where $p_t(x)$ denotes the distribution of $x$ at time $t$. To enable conditional generation, we incorporate conditional information $c$, leading to the conditional flow-matching objective (Lipman et al., 2022):

$$\mathcal{L}_{\mathrm{CFM}}(\theta) = \mathbb{E}_{t,p_1(x_1),p_t(x|x_1)} \|v(x,t|c;\theta) - u(x,t|x_1,c)\|^2. \tag{6}$$

Flow-matching proposes a straight path from noise to data. Specifically, we use a linear interpolation between the data $x_1$ and Gaussian noise $x_0$ to generate samples at time $t$:

$$x_t = (1-t)x_0 + tx_1. \tag{7}$$

Thus, the conditional vector field becomes $u(x,t|x_1,c) = x_1 - x_0$, and the rectified flow-matching (RFM) loss used for gradient descent is:

$$\|v(x,t|c;\theta) - (x_1 - x_0)\|^2. \tag{8}$$

If the vector field $u$ is estimated correctly, we can generate realistic data by propagating Gaussian noise through an ODE solver at discrete time steps. A widely used method for solving the reverse flow is the Euler ODE:

$$x_{t+\epsilon} = x + \epsilon v(x,t|c;\theta), \tag{9}$$

where $\epsilon$ is the step size. In our VocalBand, we use content, timbre, prompt style, text tokens, and other inputs for each task as conditioning information $c$, while the target data $x_1$ consists of target style, F0, or mel-spectrograms. In our AccompBand, we use timestep, text tokens, and vocal embedding as conditioning information $c$, while the target data $x_1$ is the accompaniment embedding.

Moreover, flow matching models require 100 to 1000 steps during training, but since they generate a straight path, they only require 25 or fewer steps during inference, making the generation highly efficient for fast generation. Additionally, flow-matching models ensure stable and high-quality generation due to their ability to model smooth transitions between noise and data, maintaining fidelity throughout the process. This stability is crucial for complex generation tasks, as it reduces artifacts and enhances the consistency of the output across various conditions.

# B MULTIBAND DETAILS

## B.1 MODEL DETAILS

Our MultiBand framework consists of four models: VocalBand, AccompBand, LyricBand, and MelodyBand. For the text encoder, we use FLAN-T5-large (Chung et al., 2024), while we also test BERT-large (Devlin et al., 2018) and the text encoder of CLAP (Elizalde et al., 2023) in Appendix I.1. Our vocoder is the pre-trained HiFi-GAN (Kong et al., 2020). For detailed hyperparameters of each component, please refer to Appendix C.1, D.1, E.1, and E.2.

For training details, we set the sample rate to 48kHz, the window size to 1024, the hop size to 320, and the number of mel bins to 80 to derive mel-spectrograms from raw waveforms. We train VocalBand on 4 NVIDIA RTX-4090 GPUs for 200k steps. The Adam optimizer is used with $\beta_1 = 0.9$ and $\beta_2 = 0.98$. AccompBand is trained on 8 NVIDIA RTX-4090 GPUs for 80k steps, using the AdamW optimizer with a base learning rate of $3 \times 10^{-6}$. The pre-trained Aligned Vocal and Accomp Encoder with Aligned Accomp Decoder are trained on 4 NVIDIA RTX-4090 GPUs for 40k steps. MelodyBand is trained for 30k steps until convergence on 4 NVIDIA RTX-4090 GPUs. LyricBand is fine-tuned for 4k steps until convergence on 4 NVIDIA RTX-4090 GPUs.

## B.2 TRAINING PROCEDURES

For VocalBand, the final loss terms in the training phase include the following components: 1) $\mathcal{L}_{commit}$: the commitment loss for the residual style encoder in Equation 10; 2) $\mathcal{L}_{style}$: the flow matching loss of Flow-based Style Predictor in Equation 1; 3) $\mathcal{L}_{pitch}$: the flow matching loss of Flow-based Pitch Predictor; 4) $\mathcal{L}_{mel}$: the flow matching loss of Flow-based Mel Decoder; 5) $\mathcal{L}_{dur}$: the MSE duration loss between the predicted and the GT phoneme-level duration in the log scale.

As for AccompBand, the final loss terms during training consist of the following aspects: 1) $\mathcal{L}_{balance}$: the load-balancing loss for each expert group in Band-MOE in Equation 15; 2) $\mathcal{L}_{flow}$: the flow matching loss of AccompBand.

For the pre-trained Aligned Vocal and Accomp Encoder, along with the Aligned Accomp Decoder, the final loss terms include: 1) $\mathcal{L}_{sty}$: the contrastive objective for stylistic contrast in Equation 2; 2) $\mathcal{L}_{tem}$: the contrastive objective for temporal contrast; 3) $\mathcal{L}_{rec}$: the L2 reconstruction loss; 4) $\mathcal{L}_{adv}$: the LSGAN-styled adversarial loss in GAN discriminator.

Regarding MelodyBand, the final loss terms for training involve: 1) $\mathcal{L}_{pitch}$: the cross-entropy loss for note pitches in Equation 16; 2) $\mathcal{L}_{duration}$: the L2 loss for note durations in Equation 17.

## B.3 MULTI-TASK INFERENCE PROCEDURES

If full lyrics are not provided, LyricBand generates phonemes $p$ based on the text tokens $z_p$. Without input music scores, MelodyBand generates notes $n$ (note pitches and note durations) based on lyrics, text prompts, and optional vocal prompts.

For the song generation task, VocalBand generates the target vocal $y_v$ based on $n$ and $p$ as content information, along with $z_p$ to control style information. AccompBand generates the target accompaniment $y_a$ from $y_v$ and Gaussian noise $\epsilon$.

To conduct singing style transfer, VocalBand additionally takes a vocal prompt $\tilde{y}_a$ as input to extract timbre $z_t$ and prompt style $\tilde{z}_s$. The target vocal is required to maintain consistent timbre and personal style (e.g., pronunciation, articulation skills). The Flow-based Style Predictor is used to predict the target style $z_s$, learning both personalized styles from $\tilde{z}_s$ and style control information from $z_p$ (such as singing techniques, emotions, and methods).

For music style transfer, AccompBand uses the noisy prompt accompaniment $\tilde{y}_a$ with a time step 0.5 instead of Gaussian noise $\epsilon$ and sums it with target vocal $y_v$, enabling the model to learn the style from the retained components of the prompt accompaniment.

In the vocal-to-song task, the GT vocal is used to guide AccompBand in generating the accompaniment. In contrast, for the accompaniment-to-song task, notes $n$ are extracted from the GT accompaniment $\hat{y}_a$ using ROSVOT (Li et al., 2024b) to guide VocalBand in vocal generation.

# C VOCALBAND DETAILS

## C.1 MODEL CONFIGURATION

We list the architecture and hyperparameters of VocalBand in Table 8.

Table 8: Hyper-parameters of VocalBand.

| Hyperparameter | | VocalBand |
|---|---|---|
| Phoneme Encoder | Phoneme Embedding | 256 |
| | Encoder Layers | 4 |
| | Encoder Hidden | 256 |
| | Encoder Conv1D Kernel | 9 |
| | Encoder Conv1D Filter Size | 1024 |
| Note Encoder | Pitch Embedding | 256 |
| | Duration Hidden | 256 |
| Timbre Encoder | Encoder Layers | 5 |
| | Hidden Size | 256 |
| | Conv1D Kernel | 31 |
| Residual Style Encoder | Conv Layers | 5 |
| | RQ Codebook Size | 256 |
| | Depth of RQ | 4 |
| Flow-based Style Predictor | Conv Layers | 20 |
| | Kernel Size | 3 |
| | Residual Channel | 256 |
| | Hidden Channel | 256 |
| | Training Time Steps | 100 |
| Flow-based Pitch Predictor | Conv Layers | 12 |
| | Kernel Size | 3 |
| | Residual Channel | 192 |
| | Hidden Channel | 256 |
| | Training Time Steps | 100 |
| Flow-based Mel Decoder | Conv Layers | 20 |
| | Kernel Size | 3 |
| | Residual Channel | 256 |
| | Hidden Channel | 256 |
| | Training Time Steps | 100 |
| Total Number of Parameters | | 56.26M |

## C.2 DECOMPOSITION STRATEGY

We assume that the target vocal $y_v$ can be decomposed into three distinct representations: content $z_c$, style $z_s$ (e.g., singing methods, emotion, techniques, pronunciation, and articulation skills), and timbre $z_t$. When a vocal prompt $\tilde{y}_v$ is provided during training, our goal is to transfer both the timbre $\tilde{z}_t$ and personalized style $\tilde{z}_s$ (like pronunciation and articulation skills) from the vocal prompt to the target vocal $y_v$. Meanwhile, we also need to achieve style control from text tokens $z_p$ (such as singing method, emotion, and techniques).

Following previous style transfer approaches (Jiang et al., 2024), we assume that the mutual information between $y_v$ and $\tilde{y}_v$ primarily captures global information, represented by $z_t$ (timbre). Therefore, the target timbre $z_t$ is set equal to the prompt timbre $\tilde{z}_t$, as we aim to control the timbre of the output based on the user's input. Under this assumption, $\tilde{z}_t$ is extracted using a timbre encoder, which focuses solely on timbre information, without capturing style $z_s$ or content $z_c$. To ensure that the content encoders extract only content-related information, we feed it phoneme sequences and musical notes, allowing it to exclusively pass the content representation $z_c$. For more details about the timbre encoder and the content encoders, please refer to Appendix C.3 and C.4.

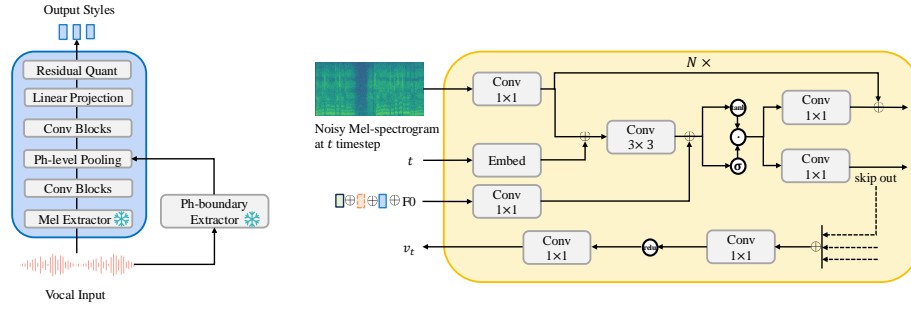

(a) Residual Style Encoder  (b) Vector Field Estimator

Figure 4: The architecture of two components of VocalBand, Figure (a) shows the residual style encoder while Figure (b) illustrates the vector field estimator of the Flow-based Mel Decoder.

Once both $z_c$ and $z_t$ are obtained, we must remove fine-grained content and timbre information from the target style $z_s$. We employ a residual style encoder to extract the prompt style $\tilde{z}_s$, and then use the Flow-based Style Predictor to predict the target style $z_s$. The latent vector $z_s$ generated by the Flow-based Style Predictor not only captures the personalized styles consistent with the prompt style $\tilde{z}_s$ (e.g., pronunciation and articulation skills) but also incorporates the styles specified in the text tokens $z_p$ (like singing methods, emotions, and techniques).

By utilizing a residual quantization (RQ) model (Lee et al., 2022a) as an information bottleneck (Qian et al., 2019), the residual style encoder is compelled to transmit only the fine-grained style information $z_s$ (Zhang et al., 2024), which other encoders cannot capture. Both $z_s$ and $\tilde{z}_s$ share the same form as the RQ embeddings, consisting of multiple layers of fine-grained style information that are disentangled from both timbre and content. This is because $z_s$ is the output of the flow-matching ODE solver, whose training objective is to capture the target style from the ground truth vocals, as extracted by the residual style encoder. For more details about the Flow-based Style Predictor, please refer to Appendix C.6. Consequently, the process guarantees the successful decomposition of style from content and timbre. These embeddings $z_c$, $z_t$, and $z_s$ are then fed into a duration predictor (Ren et al., 2020b) and a length regulator for subsequent F0 and mel-spectrogram prediction.

## C.3  TIMBRE ENCODER

The timbre encoder, designed to capture the unique identity of the singer, extracts a global timbre vector $\tilde{z}_t$ from the vocal prompt $\tilde{y}_v$. The encoder consists of multiple stacked convolutional layers. To ensure stability in the timbre representation, the output of the timbre encoder is temporally averaged, producing a one-dimensional timbre vector $\tilde{z}_t$. The target timbre $z_t$ is set equal to the prompt timbre $\tilde{z}_t$, as we aim to control the timbre of the output based on the user's input.

## C.4  CONTENT ENCODERS

Our content encoders consist of a phoneme encoder and a note encoder. The phoneme encoder processes a sequence of phonemes $p$ through a phoneme embedding layer followed by four FFT blocks, extracting phoneme features. In parallel, the note encoder handles musical score information $n$, processing note pitches and durations. These are passed through two separate embedding layers and a linear projection layer, which generate the corresponding note features. The outputs of the phoneme encoder and the note encoder are then summed as $z_c$.

## C.5  RESIDUAL STYLE ENCODER

Singing style can vary across and within phonemes. To comprehensively capture phoneme-level styles (such as singing methods, emotion, techniques, pronunciation, and articulation skills) and disentangle them from timbre and content, we design the residual style encoder. In the residual style encoder, we employ a Residual Quantization (RQ) module (Lee et al., 2022a) to extract singing style, creating an information bottleneck that effectively filters out non-style information (Zhang et al., 2024).

Thanks to the RQ's ability to extract multiple layers of information, it enables more comprehensive and detailed modeling of style across various hierarchical levels. Specifically, pronunciation and articulation skills encompass pitch transitions between musical notes and vibrato within a phoneme, where the multi-level modeling capability of RQ is highly suitable.

More concretely, as illustrated in Figure 4 (a), we first extract the mel-spectrogram from the input vocal using the open-source tool librosa [1] and further refine it through convolutional blocks. The output is then condensed into phoneme-level hidden states via a pooling layer, which operates based on phoneme boundaries. We utilize open-source tools including WhisperX (Bain et al., 2023) and Montreal Forced Aligner (MFA) (McAuliffe et al., 2017) to extract these phoneme boundaries directly from the input vocal. Subsequently, the convolution stacks capture phoneme-level correlations. Next, we use a linear projection to map the output into a low-dimensional latent variable space for code index lookup, significantly enhancing the utilization of the codebook (Yu et al., 2021).

With a quantization depth of $n$, the RQ module represents the input $z_e$ as a sequence of $N$ ordered codes. Let $RQ_i(z_e)$ denote the process of representing $z_e$ as RQ code and extracting the code embedding in the $i$-th codebook. The representation of $z_e$ in the RQ module at depth $n \in [N]$ is denoted as $\hat{z_e}^n = \sum_{i=1}^{n} RQ_i(z_e)$. To ensure that the input representation adheres to a discrete embedding, a commitment loss (Lee et al., 2022a) is employed:

$$\mathcal{L}_{commit} = \sum_{n=1}^{N} \|z_e - sg[\hat{z_e}^n]\|_2^2, \tag{10}$$

where the notation $sg$ represents the stop-gradient operator. It is important to note that $\mathcal{L}_{commit}$ is the cumulative sum of quantization errors across all $n$ iterations, rather than a single term. The objective is to ensure that $\hat{z_e}^n$ progressively reduces the quantization error of $z_e$ as the value of $n$ increases. Finally, we extract the phoneme-level style embedding from the input vocal.

## C.6 FLOW-BASED STYLE PREDICTOR

As shown in Figure 3 (a), the Flow-based Style Predictor uses content $z_c$, timbre $z_t$, phoneme-level prompt style $\tilde{z_s}$, and text tokens $z_p$ to predict the target style $z_s$. With the combined $z_c$ and $z_t$, we employ a style alignment model utilizing the Scaled Dot-Product Attention mechanism (Vaswani et al., 2017) to align style control information from $z_p$ (e.g., singing methods, emotions, techniques) with the content. Positional embedding is applied before feeding $z_p$ into the attention module. In the attention module, the combined $z_c$ and $z_t$ serve as the query $z_{ct}$, while $z_p$ serves as both the key and value, and $d$ represents the dimensionality of the key and query:

$$Attention(Q, K, V) = Attention(z_{ct}, z_p, z_p) = Softmax\left(\frac{z_{ct} z_p^T}{\sqrt{d}}\right) z_p. \tag{11}$$

We stack the style alignment layer multiple times for better performance and gradually stylize the query value. We combine the output with $z_{ct}$ as condition $c$ and then feed it into an ODE solver, which transforms Gaussian noise $\epsilon$ into $z_s$ along a probability path $p_t(z_{st})$. We concatenate $\tilde{z_s}$ with $\epsilon$ to allow $z_s$ to learn personalized styles (e.g., pronunciation and articulation skills).

During training, we set $u(z_{st}, t)$ to represent the target vector field at time $t$, obtained through linear interpolation between $\epsilon$ and the ground truth (GT) phoneme-level style $z_s$, which is extracted from the GT vocal by the residual style encoder. To stabilize the flow-matching training process, we do not train the Flow-based Style Predictor during the early stages of training (the first 50,000 steps). Instead, we feed the GT style $z_s$ into the subsequent Flow-based Pitch Predictor and Mel Decoder. Therefore, by the time we begin training the Flow-based Style Predictor, the residual style encoder has stabilized, ensuring a consistent GT $z_s$, which is beneficial for the flow-matching training.

The learned vector field $v(z_{st}, t|c; \theta)$, predicted by a vector field estimator at each time $t$, ensures smooth interpolation between the initial noise and the output $z_s$, guided by the flow-matching objective. We use the non-causal WaveNet architecture (Van Den Oord et al., 2016) as the backbone of our vector field estimator, due to its proven capability in modeling sequential data. For more details about the vector field estimator, please refer to Appendix C.8. Notably, to enable the model to handle

---

[1] https://github.com/librosa/librosa

cases without a vocal prompt, we drop vocal prompts with a probability of 0.2 during training. We also replace $z_p$ with embedded empty strings in a probability of 0.1 for cases without prompts.

During inference, the ODE solver generates the phoneme-level target style $z_s$ directly from the concatenation of Gaussian noise and $\tilde{z}_s$ (if a vocal prompt is provided), based on the condition $c$. This method ensures fast and controllable generation of $z_s$, learning personalized styles consistent with $\tilde{z}_s$ while incorporating the aligned style control information from $z_p$.

### C.7 FLOW-BASED PITCH PREDICTOR AND MEL DECODER

During training, our target F0 is extracted using the open-source tool RMVPE (Wei et al., 2023), while mel-spectrograms are extracted using the open-source tool librosa [1]. We adopt the non-causal WaveNet architecture (Van Den Oord et al., 2016) as the backbone of our vector field estimator. For further details on the vector field estimator, please refer to Appendix C.8.

### C.8 VECTOR FIELD ESTIMATOR

We adopt the non-causal WaveNet architecture (Van Den Oord et al., 2016) as the backbone of our vector field estimators for the Flow-based Style Predictor, Pitch Predictor, and Mel Decoder, due to its demonstrated effectiveness in modeling sequential data. The architecture of the vector field estimator for the Flow-based Mel Decoder is depicted in Figure 4 (b). We input content $z_c$, timbre $z_t$, style $z_s$, and F0 as conditioning factors to predict the corresponding vector field. Similarly, the architecture of the vector field estimators for the Flow-based Pitch Predictor and Style Predictor follows the same structure, while the only difference lies in the input and condition for each model.

## D ACCOMPBAND DETAILS

### D.1 MODEL CONFIGURATION

We list the architecture and hyperparameters of AccompBand in Table 9.

Table 9: Hyper-parameters of AccompBand.

| Hyperparameter | | AccompBand |
|---|---|---|
| Aligned Vocal Encoder | Encoder Layers | 3 |
| | Encoder Hidden | 384 |
| | Encoder Conv1D Kernel | 5 |
| | Encoder Output Channels | 20 |
| Aligned Accomp Encoder | Encoder Layers | 3 |
| | Encoder Hidden | 384 |
| | Encoder Conv1D Kernel | 5 |
| | Encoder Output Channels | 20 |
| Aligned Vocal Encoder | Decoder Layers | 3 |
| | Decoder Hidden | 384 |
| | Decoder Conv1D Kernel | 5 |
| | Decoder Input Channels | 20 |
| Band Transformer Blocks | Transformer Layers | 4 |
| | Transformer Embed Dim | 768 |
| | Transformer Attention Headers | 8 |
| | Experts for each group | 4 |
| | Training Time Steps | 1000 |
| Total Number of Parameters | | 431.07M |

### D.2 ALIGNED VOCAL AND ACCOMP ENCODER

For training the Aligned Vocal and Accompaniment Encoder, we use the contrastive objective (Radford et al., 2021), and design two types of objectives: $\mathcal{L}_{sty}$ and $\mathcal{L}_{tem}$. For $\mathcal{L}_{sty}$, we maximize

the similarity of vocal-accompaniment pairs from the same song while minimizing the similarity for vocal-accompaniment pairs from different songs. During sample selection, we randomly sample multiple negative samples from different songs. For $\mathcal{L}_{tem}$, we maximize the similarity of vocal-accompaniment pairs from the same time segment within a song and minimize the similarity for pairs from different time segments of the same song. In this case, we randomly sample multiple negative samples from different segments of the same song.

For training the Aligned Accompaniment Decoder, we use the L2 reconstruction loss: $\mathcal{L}_{rec} = \|y_v - \hat{y_v}\|^2$, where $y_v$ is the reconstructed accompaniment mel-spectrogram and $\hat{y_v}$ is the ground truth accompaniment mel-spectrogram. Additionally, we incorporate a GAN discriminator, following the architecture of ML-GAN (Chen et al., 2020), to further enhance the quality of the reconstruction. We apply the LSGAN-style adversarial loss (Mao et al., 2017), $\mathcal{L}_{adv}$, which aims to minimize the distributional distance between the predicted mel-spectrograms and the ground truth mel-spectrograms. Before feeding the waveform into these encoders, we first extract the mel-spectrogram using librosa [1]. After generating the mel-spectrogram from the decoder output, we utilize HiFi-GAN (Kong et al., 2020) to convert it back into audio.

### D.3 BAND TRANSFORMER BLOCKS

As shown in Figure 2 (c), the Band Transformer Blocks are based on Flag-Dit (Gao et al., 2024). During training, the vocal embedding $z_v$ extracted by the Aligned Vocal Encoder is added to the noisy input $x_t$ to leverage the transformer's self-attention mechanism, allowing the model to learn vocal-matching style, rhythm, and melody. We use RMSNorm (Zhang & Sennrich, 2019) to improve training stability, preventing the absolute values from growing uncontrollably and causing numerical instability. Next, we compute the global style embedding $z_g$ by averaging the text tokens $z_p$ and vocal embedding $z_v$ along the temporal dimension and adding the time step embedding of $t$. This global style embedding is used in a multi-layer style adaptor, which modulates the latent representation using adaptive layer normalization (AdaLN) (Peebles & Xie, 2023) to ensure style consistency. We compute the scale and shift using linear regression based on $z_g$:

$$AdaLN(h, c) = \gamma_c \times LayerNorm(h) + \beta_c, \tag{12}$$

where $h$ represents the hidden representation. We zero-initialize the batch norm scale factor $\gamma$ in each block (Peebles & Xie, 2023). Moreover, we explore relative positional encoding with rotary positional embedding (RoPE) (Su et al., 2024), which injects temporal positional information into the model. This enables the model to capture the temporal relationships between successive frames, providing significant performance improvements for the transformer.

Then, the zero-initialized attention mechanism (Bachlechner et al., 2021) is used to inject conditional information from the text tokens $z_p$ into the hidden states $h$, while simultaneously learning the vocal style, rhythm, and melody aligned with the vocal embedding $z_v$ added to $x_t$. Given the accompaniment queries $Q_h$, keys $K_h$, and values $V_h$ from hidden states, along with the text keys $K_z$ and values $V_z$, the final attention output is formulated as:

$$Attention = softmax\left(\frac{\tilde{Q}_h \tilde{K}_h^\top}{\sqrt{d}}\right) V_h + \tanh(\alpha) softmax\left(\frac{\tilde{Q}_h K_z^\top}{\sqrt{d}}\right) V_z, \tag{13}$$

where $\tilde{Q}_h$ and $\tilde{K}_h$ denote using RoPE in queries and keys, $d$ is the dimensionality of queries and keys, and $\alpha$ is a zero-initialized learnable parameter that gates the cross-attention with the text tokens.

### D.4 BAND-MOE

As illustrated in Figure 3(d), Band-MOE is composed of three expert groups: Aligned MOE, Controlled MOE, and Acoustic MOE, each comprising multiple experts. We employ Feed-Forward Networks (FFNs) as the architecture for each expert. It is well-established (Lee et al., 2022b; Huang et al., 2022b) that mel-spectrogram details exhibit different patterns across various acoustic frequencies. In musical accompaniment, high-frequency components often include the harmonics and overtones of instruments like strings and flutes, as well as percussive elements such as cymbals and hi-hats, which enhance the brightness and clarity of the sound. Conversely, low-frequency content encompasses basslines and kick drums, providing foundational rhythm and depth that shape the

overall groove and warmth of the music. Motivated by this, previous works (Kong et al., 2020; Yang et al., 2021) have adopted multi-scale architectures to model downsampled signals at different frequency bands, which effectively control the periodic elements of the signal and reduce artifacts.

Building on this idea, we introduce Acoustic MOE, where experts are assigned to specific acoustic frequency bands based on the processed hidden representation $h$, and their outputs are aggregated to produce the final result. Moreover, since the Aligned Vocal and Accomp Encodequ di ar employ 1D convolutions to encode both the vocal and accompaniment mel-spectrograms, the latent representation of the hidden $h$ should retain the acoustic frequency distribution.

Our routing strategy for all routers is based on the dense-to-sparse Gumbel-Softmax method (Nie et al., 2021), enabling dynamic and efficient expert selection. The Gumbel-Softmax trick facilitates sampling from a categorical distribution by reparameterizing categorical variables to make them differentiable. Specifically, the routing score $g(h)$ for each expert $i$ is computed as follows:

$$g(h)_i = \frac{\exp((h \cdot W_g + \zeta_i)/\tau)}{\sum_{j=1}^{N} \exp((h \cdot W_g + \zeta_j)/\tau)}, \tag{14}$$

where $W_g$ is the learned gating weight, $\zeta$ is sampled from the Gumbel(0, 1) distribution (Jang et al., 2016), and $\tau$ is the softmax temperature. Initially, a high temperature $\tau$ results in denser expert selection, allowing multiple experts to process the same input. As training progresses, $\tau$ is gradually decreased, making the routing sparser and selecting fewer experts for each input. When $\tau \to 0$, the distribution approaches a nearly one-hot form, effectively selecting the most suitable expert for each token. Following prior work (Nie et al., 2021), we dynamically reduce $\tau$ from 2.0 to 0.3 during training and use the hard mode during inference, selecting only one expert. Notably, only the global router does not conduct hard mode during inference, as we need experts from different expert groups to cooperate in accompaniment generation. The algorithm of Band-MOE is shown in Algorithm 1.

---

**Algorithm 1** Pseudo-Code of Band-MOE Routing Strategy

---

**Input:** Input hidden representation $h$, vocal embedding $z_v$, text prompt embedding $z_p$, time step $t$
**Output:** Output with enhanced quality and control $o_{\text{final}}$
  1: Initialize Gumbel-Softmax temperature $\tau$, sample Gumbel noise $\zeta$
  2: **for** each time step $t$ **do**
  3:     **Aligned MOE:**
  4:         Use Gumbel-Softmax for each token in the time channel to select an expert by $z_v$:
  5:         $g_{\text{aligned}}(h) \leftarrow \text{GumbelSoftmax}(z_v \cdot W_{\text{aligned}} + \zeta)/\tau$
  6:         Compute Aligned MOE output:
  7:         $o_{\text{aligned}} \leftarrow \sum_i g_{\text{aligned},i} \cdot \text{Expert}_{i,\text{aligned}}(z_v)$
  8:     **Controlled MOE:**
  9:         Use Cross-Attention extracting style for alignment between $z_p$ and $h$:
 10:         $z_{sty} \leftarrow \text{CrossAttention}(h(Q), z_p(K), z_p(V))$
 11:         Use Gumbel-Softmax for each token in the time channel to select an expert by $z_{sty}$:
 12:         $g_{\text{controlled}}(h) \leftarrow \text{GumbelSoftmax}(z_{sty} \cdot W_{\text{controlled}} + \zeta)/\tau$
 13:         Compute Controlled MOE output:
 14:         $o_{\text{controlled}} \leftarrow \sum_i g_{\text{controlled},i} \cdot \text{Expert}_{i,\text{controlled}}(z_p)$
 15:     **Global Router:**
 16:         Use Gumbel-Softmax to compute global weights $\alpha_t$ and $\beta_t$:
 17:         $g_{\text{global}}(h) \leftarrow \text{GumbelSoftmax}(embedding(t) \cdot W_{\text{global}} + \zeta)/\tau$
 18:         $\alpha_t, \beta_t \leftarrow g_{\text{global}}(h)$
 19:         Combine Aligned and Controlled MOE outputs:
 20:         $o_{\text{combined}} \leftarrow \alpha_t \cdot o_{\text{aligned}} + \beta_t \cdot o_{\text{controlled}}$
 21:     **Acoustic MOE:**
 22:         Use Gumbel-Softmax to select an expert for each frequency channel:
 23:         $g_{\text{acoustic}}(o_{\text{combined}}) \leftarrow \text{GumbelSoftmax}(o_{\text{combined}} \cdot W_{\text{acousitc}} + \zeta)/\tau$
 24:         Compute Acoustic MOE output:
 25:         $o_{\text{acoustic}} \leftarrow \sum_j g_{\text{acoustic},j} \cdot \text{Expert}_{j,\text{acoustic}}(o_{\text{combined}})$
 26: **end for**
 27: Return $o_{\text{final}} \leftarrow o_{\text{acoustic}}$ as the final routed output

---

Moreover, to avoid overloading any individual expert and ensure balanced utilization, we incorporate a load-balancing loss for each expert group (Fedus et al., 2022). The balance loss $\mathcal{L}_{balance}$ is:

$$\mathcal{L}_{balance} = \alpha N \sum_{i=1}^{N} \left( \frac{1}{B} \sum_{h \in B} g(h)_i \right). \tag{15}$$

where $B$ is the batch size, $N$ is the number of experts, and $\alpha$ is a hyperparameter controlling the strength of the regularization, for which we use 0.1. This loss encourages a more uniform distribution of tokens across experts, improving training efficiency by preventing expert underutilization or overload. Thus, our routing strategy not only allows dynamic expert selection but also ensures that the computational load is evenly distributed across experts, reducing training time and improving the model performance of Band-MOE.

### D.5 Classifier-free Guidance

During AccompBand training, we randomly replace the input text tokens with embedded empty strings at a probability of 0.2. The empty strings, like the original text prompts, are processed through the text encoder to extract text tokens and are padded to a fixed length. For $\gamma$ in Equation 3, a higher $\gamma$ emphasizes the control of the text prompt, improving generation quality by making the outputs more aligned with the given conditions. In contrast, a lower $\gamma$ allows for more diverse outputs by reducing the reliance on the text prompt, though this may result in lower relevance to the input prompt. In our major accompaniment generation experiments, we use $\gamma = 3$.

## E LyricBand and MelodyBand

### E.1 LyricBand

To enhance the customizability of our song generation system, we introduce LyricBand, a model designed to generate complete song lyrics based on arbitrary text prompts. Users can input parameters such as theme, emotion, genre, style, and specific keywords to generate fully personalized lyrics tailored to their preferences. To effectively train LyricBand, we leverage GPT-4o (Achiam et al., 2023) to extract prompts from a large corpus of existing song lyrics in our training data. These prompts encapsulate essential elements such as the thematic content, emotional tone, narrative perspective, rhyme scheme, and stylistic features of the songs. By extracting this rich set of attributes, we create a comprehensive dataset that pairs textual prompts with corresponding lyrics, enabling the model to learn the mapping between user inputs and desired lyrical outputs.

We employ QLoRA (Dettmers et al., 2024) for efficient fine-tuning of the well-performing open-source bilingual large language model Qwen-7B (Bai et al., 2023). By utilizing 4-bit quantization and low-rank adapters, QLoRA significantly reduces the computational resources required for fine-tuning while preserving the model's performance. This approach allows LyricBand to adapt effectively to the task of lyrics generation, maintaining high levels of customization and creativity across a diverse range of user prompts. In our experiments, we set LoRA $r = 32, \alpha = 16$. LyricBand demonstrates the capability to capture nuanced themes and emotions specified by users, generating lyrics that not only align with the given prompts but also exhibit coherent structure and artistic expression.

### E.2 MelodyBand

Previous singing voice and song generation models often require users to provide music scores to achieve stable melodies (Zhiqing et al., 2024), lacking personalized customization of the melody. Inspired by symbolic music generation models (Dong et al., 2018; Ding et al., 2024), we introduce MelodyBand, an additional model where melody-related features like notes are generated from text descriptions in advance. By using notes as the representation of the melody, we can achieve more stable melody control. However, requiring users to provide music scores is impractical. Generating notes using natural language prompts can both ensure stable melodies and allow for flexible customization. For controllable melody generation, we construct artificial textual prompts to deliver melody-related information. Musical attributes like key, tempo, vocal range, and other information can be used as prompts for melody customization.

When users do not input music scores, as shown in Figure 5(b), MelodyBand takes the phonemes of the lyrics as content information and optional vocal prompts to extract timbre. It composes music for the lyrics and selects appropriate frequencies based on the timbre, using text prompts for style control. We employ a non-autoregressive transformer model to efficiently generate note pitches and durations simultaneously. The non-autoregressive transformer enables fast and high-quality generation, making it highly suitable for our multi-task song generation system.

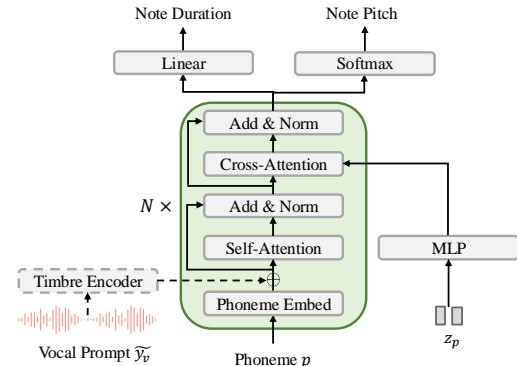

Figure 5: The architecture of MelodyBand.

With encoded phonemes and timbre, we inject text prompts through cross-attention transformers, allowing the model to integrate linguistic cues more effectively. Several heads are added to generate note pitches and durations. We pass each dimension of the stacked output through a softmax function to generate note pitches and through a linear layer to generate note durations. We train MelodyBand using cross-entropy loss for note pitches and an L2 loss for note durations. Let the true note pitch and duration for $i$-th phoneme be $n_p^{(i)}$ and $n_d^{(i)}$, and the GT note pitch and duration be $\hat{n}_p^{(i)}$ and $\hat{n}_d^{(i)}$, respectively. The cross-entropy loss $\mathcal{L}_{pitch}$ is:

$$\mathcal{L}_{pitch} = -\sum_{i=1}^{N}\sum_{k=1}^{K} \delta_{\hat{n}_p^{(i)},k} \log(P_k^{(i)}), \tag{16}$$

where $N$ is the length of phoneme sequence, $K$ is number of pitch classes, $\delta_{\hat{n}_p^{(i)},k}$ is 1 if $\hat{n}_p^{(i)} = k$ and 0 otherwise, and $P_k^{(i)}$ is the predicted probability of pitch $k$ at time $i$. The L2 loss $\mathcal{L}_{duration}$ is:

$$\mathcal{L}_{duration} = \sum_{i=1}^{N} \left( n_d^{(i)} - \hat{n}_d^{(i)} \right)^2. \tag{17}$$

Our MelodyBand employs 8 transformer layers, and 8 attention heads, the hidden size is 768, with 23.32M parameters in total.

## F    DATASET ANALYSIS

Table 10: Statistics of training datasets.

| Dataset | Type | Languages | Annotation | Duration (hours) |
|---|---|---|---|---|
| Opencpop (Wang et al., 2022) | vocal | Chinese | lyrics, notes | 5.3 |
| M4Singer (Zhang et al., 2022a) | vocal | Chinese | lyrics, notes | 29.8 |
| OpenSinger (Huang et al., 2021) | vocal | Chinese | lyrics | 83.5 |
| PopBuTFy (Liu et al., 2022a) | vocal | English | lyrics | 10.8 |
| LP-MusicCaps-MSD (Doh et al., 2023) | accomp | / | text prompt | 213.6 |
| web-crawled | song | Chinese, English | / | 979.4 |

We train our model using a combination of bilingual web-crawled song datasets and open-source singing datasets. Since there are no publicly available annotated song datasets, we collect 20k Chinese and English songs from well-known music websites. The open-source singing datasets we utilize are OpenCpop (Wang et al., 2022) (5 hours in Chinese), M4Singer (Zhang et al., 2022a) (30 hours in Chinese), OpenSinger (Huang et al., 2021) (83 hours in Chinese), and a subset of PopBuTFy (Liu et al., 2022a) (10 hours in English). After preprocessing and cleaning, we have approximately 1,000 hours (about 80% in Chinese and 20% in English) of song data and 1,100 hours of vocal data. For accompaniment generation, we use a filtered subset of LP-MusicCaps-MSD (Doh et al., 2023), resulting in a total size of around 1,200 hours. We use all open-source datasets under license CC BY-NC-SA 4.0. The statistics of the datasets are listed in 10.

For the web-crawled data, we use Ultimate Vocal Remover [2], an open-source music source separation tool, to perform the vocal-accompaniment separation. We utilize WhisperX (Bain et al., 2023) to automatically transcribe the demixed vocals, and Montreal Forced Aligner (MFA) (McAuliffe et al., 2017) is employed for phoneme and vocal alignment. After that, we filter the samples using Silero VAD (Team, 2021) to eliminate unvoiced clips. The samples are segmented into phrases with a maximum length of 20 seconds, resulting in an average segment duration of 12 seconds.

We utilize a music captioning model (Doh et al., 2023) to generate text prompts from the segmented song clips, and GPT-4o (Achiam et al., 2023) is used to separate music styles (such as genre, tone, and instrumentation) from vocal descriptions (such as emotion and gender). For singing styles, we hire music experts to annotate all songs for the global singing method (e.g., pop or bel canto) and to label around 200 hours of segmented vocal clips for specific techniques used. We hire all music experts and annotators with musical backgrounds at a rate of $300 per hour. They have agreed to make their contributions available for research purposes. These annotations, along with the separated vocal descriptions, form the complete singing styles. For melody styles, we extract the key from the segmented demixed vocal clips using music21 [3], tempo and duration using librosa [1], and then use GPT-4o to combine these elements, generating natural language descriptions of vocal ranges based on the average pitch. For lyric styles, we process the lyrics using GPT-4o to extract essential elements such as thematic content, emotional tone, narrative perspective, rhyme scheme, and stylistic features.

All styles are combined, along with annotations for various tasks, to form the final text prompts. During generation, we randomly omit certain elements or entire styles to enhance the model's generalization ability. We utilize ROSVOT (Li et al., 2024b) to obtain note sequences from the segmented demixed vocal clips. For vocal and accompaniment data that lacks specific annotations, we use corresponding methods to complete the labeling process.

## G  EVALUATION METRICS

### G.1  LYRIC AND MELODY EVALUATION

For lyric generation, we randomly select 30 prompts and generate 30 sets of lyrics. Each set is evaluated by at least 15 raters for overall quality (OVL) and relevance to the prompt (REL) as subjective evaluation metrics. The rating scale ranged from 1 to 100, representing poor to good quality. OVL focused on the overall quality of the lyrics, including naturalness, and grammatical correctness, while REL assessed the alignment with the thematic content, emotional tone, narrative perspective, rhyme scheme, and stylistic features specified in the text prompt. All participants are fairly compensated for their time and effort at a rate of $12 per hour. They are also informed that the results will be used for scientific research purposes. The testing screenshot is shown in Figure 6.

Figure 6: Screenshot of lyric evaluation.

In melody generation, multiple objective metrics are employed to evaluate controllability. We use the Krumhansl-Schmuckler algorithm to predict the potential key of the generated notes and report the average key accuracy (KA). If the Pearson correlation coefficient of the ground truth (GT) notes

---

[2]https://github.com/Anjok07/ultimatevocalremovergui
[3]https://github.com/cuthbertLab/music21

corresponding to the GT key is $r$, and the predicted MIDI corresponding to the GT key is $\hat{r}$, we define the key accuracy as $KA = \hat{r}/r$ (only valid if $r \neq 0$). We also compute the average absolute difference of average pitches (APD) and temporal duration (TD, in seconds). Moreover, following previous work (Sheng et al., 2021), we record pitch and duration distribution similarity (PD and DD). Specifically, we calculate the distribution (frequency histogram) of pitches and durations in notes and measure the distribution similarity between generated notes and ground truth notes:

$$\frac{1}{N_s} \sum_{i=1}^{N_s} OA(Dis_i, \hat{Dis_i}), \tag{18}$$

where $Dis_i$ and $\hat{Dis_i}$ represent the pitch or duration distribution of the $i$-th generated and ground-truth song, respectively, $N_s$ is the number of songs in the test set, and OA represents the average overlapped area. Melody distance (MD) is also computed with dynamic time warping (DTW) (Berndt & Clifford, 1994). To evaluate the pitch trend of the melody, we spread out the notes into a time series of pitch according to the duration, with a granularity of 1/16 note. Each pitch is normalized by subtracting the average pitch of the entire sequence. To measure the similarity between generated and ground-truth time series with different lengths, we use DTW to compute their distance.

## G.2 VOCAL EVALUATION

For vocal generation, we randomly select 30 pairs of sentences from our test set for subjective evaluation. Each pair consists of a ground truth (GT) and a synthesized vocal, each listened to by at least 15 professional listeners. For MOS-Q evaluations, these listeners are instructed to focus on synthesis quality (including clarity, naturalness, and richness of stylistic details) without considering the style control relevance to text prompts. For MOS-C, the listeners are informed to evaluate style controllability (relevance to the text prompt regarding the singing method, emotion, and techniques), disregarding any differences in content, timbre, or synthesis quality (such as clarity, naturalness, and stylistic details). In both MOS-Q and MOS-C evaluations, listeners are asked to grade various vocal samples on a Likert scale from 1 to 5. For fairness, all samples are resampled to 48kHZ. The screenshots of instructions for testers are shown in Figure 7.

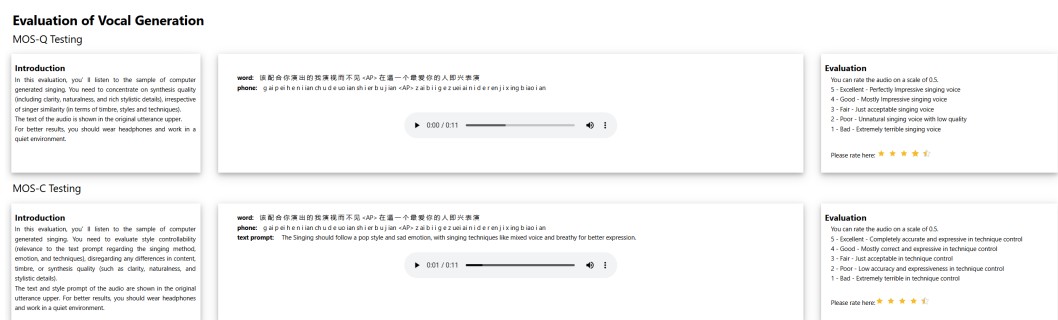

Figure 7: Screenshot of vocal evaluation.

We employ F0 Frame Error (FFE) to evaluate the test set's synthesis quality objectively. FFE combines metrics for voicing decision error and F0 error, capturing essential synthesis quality information. For comparison with FFE reported in the MelodyLM paper, we resample all audio to 24kHz for FFE.

For singing style transfer, subjective evaluation is conducted using pairs of audio, where each pair includes a prompt vocal and a synthesized vocal. During MOS-S evaluations, listeners are asked to assess singer similarity in terms of timbre and personalized styles to the vocal prompt, disregarding any differences in content or synthesis quality.

To objectively evaluate timbre similarity, we employ Cosine Similarity (Cos). Cos measures the resemblance in singer identity between the synthesized vocal and the vocal prompt by computing the average cosine similarity between the embeddings extracted from the synthesized voices and the vocal prompt, thus providing an objective indication of singer similarity performance. Specifically, we use a voice encoder [4] to extract singer embeddings.

---

[4]https://github.com/resemble-ai/Resemblyzer

In all MOS-Q, MOS-S, and MOS-C evaluations, listeners are requested to grade the vocal samples on a Likert scale ranging from 1 to 5. All participants are fairly compensated for their time and effort. We compensate participants at a rate of $12 per hour. Participants are informed that the results will be used for scientific research.

### G.3 ACCOMPANIMENT AND SONG EVALUATION

For the subjective evaluation of accompaniment and song generation, we randomly select 30 audio samples from our test set. Each sample is listened to by at least 15 raters. Following previous work (Copet et al., 2024; Zhiqing et al., 2024), we ask human raters to evaluate three aspects of the audio samples: (i) overall quality (OVL), (ii) relevance to the text prompts (REL), and (iii) alignment with the vocal (ALI). For the overall quality test, raters are asked to rate the perceptual quality of the provided samples. For the text relevance test, raters evaluate how well the audio matches the music style control information in the text prompts. For the alignment with the vocal test, raters focus on the temporal correspondence between accompaniment and vocal in terms of style, melody, and rhythm. Ratings are given on a scale from 1 to 100.

All participants are fairly compensated for their time and effort, with a rate of $12 per hour. Participants are informed that the results will be used for scientific research. For fairness, all samples are resampled to 48kHZ and normalized to -23dB LUFS (Series, 2011). The screenshots of instructions in the song generation task for testers are shown in Figure 8.

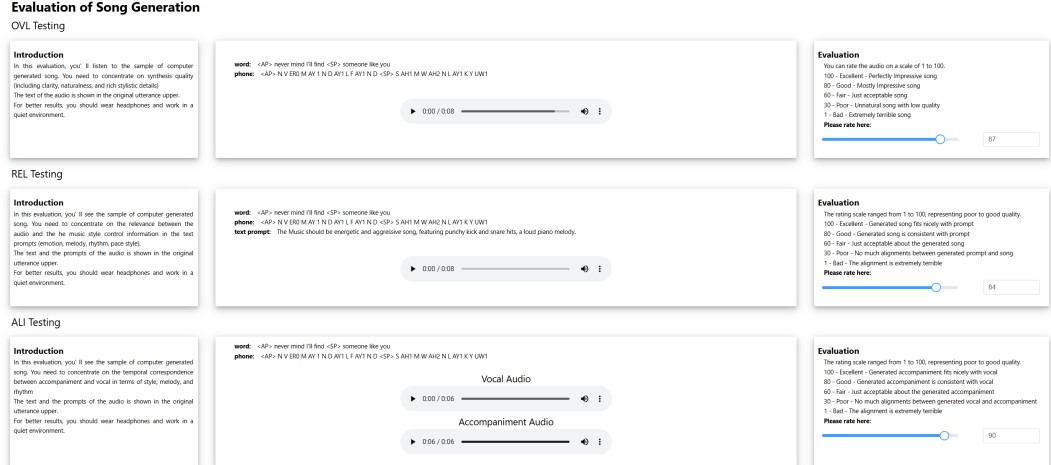

Figure 8: Screenshot of song evaluation.

For the objective evaluation, we use Frechet Audio Distance (FAD), Kullback-Leibler Divergence (KLD), and the CLAP score (CLAP). We report the FAD (Kilgour et al., 2018) using the official implementation in TensorFlow with the VGGish model [5]. A low FAD score indicates that the generated audio is plausible. Following previous work (Copet et al., 2024), we compute the KL-divergence over the probabilities of the labels between the GT and the generated music. Finally, the CLAP score (Wu et al., 2023) is computed between the track description and the generated audio to quantify audio-text alignment, using the pre-trained CLAP model [6].

## H MULTI-TASK EXPERIMENTS

### H.1 VOCAL GENERATION

In Figure 9, we compare the mel-spectrogram representations of VocalBand with different singing styles specified in the text prompt. Figure 9 (a) represents the GT vocal, where the mel-spectrogram within the yellow box is relatively uniform, indicating a stable vocal performance, while the F0

---

[5]https://github.com/google-research/google-research/tree/master/frechet_audio_distance
[6]https://github.com/LAION-AI/CLAP

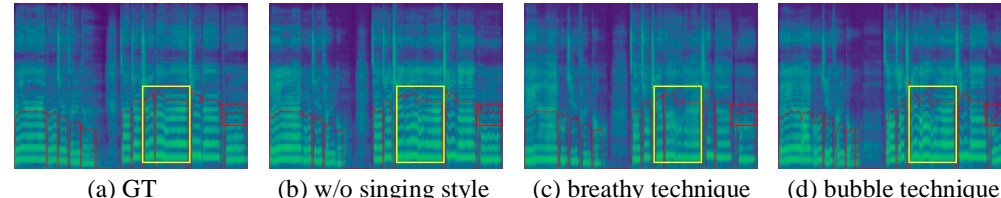

| (a) GT | (b) w/o singing style | (c) breathy technique | (d) bubble technique |

Figure 9: Visualization of the mel-spectrogram results generated by VocalBand under different singing styles in the text prompt. The red box contains the fundamental pitch, and the yellow box contains the details of harmonics.

contour in the red box is smooth. In contrast, Figure 9 (b) does not specify singing styles, allowing the free use of techniques to enhance expressiveness, as seen by the significant pitch oscillations in the red box, characteristic of vibrato. In Figure 9 (c), representing the breathy technique, the mel energy in the yellow box shows a significant drop in high-frequency energy, consistent with the softer, airier vocal timbre of breathy singing. Finally, Figure 9 (d) illustrates the bubble technique, where the yellow box displays pronounced low-frequency energy with more exaggerated vertical modulations. The red box shows a distinctive pitch fluctuation pattern, characterized by slower, larger oscillations, indicative of the unique vocal fold vibrations in this technique. These results demonstrate that VocalBand can achieve diverse and highly expressive control over the same content based on the different singing styles specified in the text prompt.

## H.2 SINGING STYLE TRANSFER

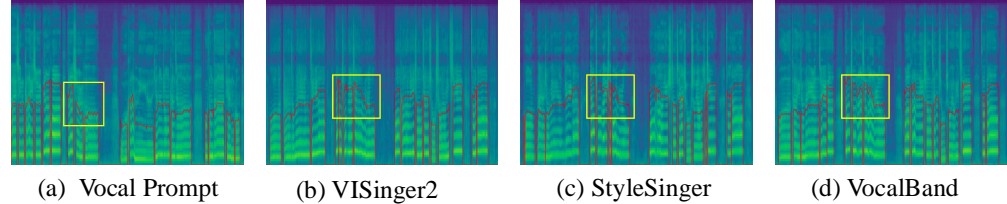

| (a) Vocal Prompt | (b) VISinger2 | (c) StyleSinger | (d) VocalBand |

Figure 10: Visualization of the mel-spectrogram results generated by VocalBand for singing style transfer. The yellow box contains the fundamental pitch.

In Figure 10, we compare the performance of VocalBand and baseline models on singing style transfer. It can be observed that VocalBand excels at capturing the intricate nuances of the prompt style. The pitch curve generated by VocalBand displays a greater range of variations and finer details, closely resembling the prompt style. In the yellow boxes, it is evident that VocalBand captures nuances in pronunciation and articulation skills similar to the vocal prompt. In contrast, the curves generated by other methods appear relatively flat, lacking distinctions in singing styles. Moreover, the mel-spectrograms generated by VocalBand exhibit superior quality, showcasing rich details in frequency bins between adjacent harmonics and high-frequency components. In contrast, the mel-spectrograms produced by other methods demonstrate lower quality and a lack of intricate details.

## H.3 MUSIC STYLE TRANSFER

Table 11: Results of music style transfer. Prompt means prompt accompaniment.

| Methods | FAD ↓ | KLD ↓ | OVL ↑ | ALI-A ↑ |
|---|---|---|---|---|
| AccompBand (w/o prompt) | 2.92 | **1.22** | **88.65±1.45** | - |
| AccompBand | **2.92** | 1.23 | 88.34±1.28 | **80.24±1.57** |

For music style transfer, AccompBand uses the noisy prompt accompaniment $\tilde{y}_a$ with a time step 0.5 instead of Gaussian noise $\epsilon$ and sums it with the target vocal $y_v$, enabling the model to learn the style

from the retained components of the prompt accompaniment. Thus, we do not need a text prompt to control the music style. We use ALI-A for subjective evaluation of the style similarity to the prompt accompaniment. As shown in Table 11, we achieve good style similarity with minimal changes in quality. This demonstrates that MultiBand, leveraging AccompBand's flow matching mechanism, can also effectively perform the music style transfer task.

## H.4 VOCAL-TO-SONG GENERATION

Table 12: Results of vocal-to-song generation. GT means GT vocal.

| Methods | FAD ↓ | KLD ↓ | OVL ↑ | ALI ↑ |
|---|---|---|---|---|
| AccompBand (w/o GT) | 2.92 | 1.22 | 88.65±1.45 | 80.72±1.49 |
| MelodyLM | 3.13 | 1.31 | 84.67±1.23 | 75.19±0.82 |
| AccompBand | **2.65** | **1.19** | **90.17±1.55** | **83.54±1.32** |

We can directly input GT vocals for the vocal-to-song generation task. We compare our method with MelodyLM, which also generates songs from GT vocals. We use the objective metrics reported in their papers and subjectively evaluate the demos on their demo pages. As shown in Table 12, it is evident that with GT vocal input, AccompBand achieves improved quality and better alignment with the vocals compared to song generation without GT vocal input, and it outperforms MelodyLM. This is because the GT vocal provides a more accurate style, melody, and rhythm, better matching the target accompaniment. It demonstrates that MultiBand effectively utilizes AccompBand's excellent vocal alignment mechanisms, including the Aligned Vocal Encoder and Aligned MOE, to accomplish the Vocal-to-Song Generation task.

## H.5 ACCOMPANIMENT-TO-SONG GENERATION

We use ROSVOT (Li et al., 2024b) to extract notes from the accompaniment to guide VocalBand for vocal generation. The extracted notes are also provided to StyleSinger, which can similarly utilize notes, as a baseline model. As shown in Table 13, it is evident that the quality decreases when using GT accompaniment instead of music scores, as the notes from the accompaniment are not aligned with the vocal notes, primarily due to differences in their characteristics. Vocals often involve techniques and emotional expression, with pauses between words. At the same

Table 13: Results of accompaniment-to-song generation. GT means GT accompaniment.

| Methods | MOS-Q ↑ | FEE ↓ |
|---|---|---|
| VocalBand (w/o GT) | 4.04±0.08 | 0.07 |
| StyleSinger | 3.79±0.10 | 0.09 |
| VocalBand | **3.87±0.05** | **0.08** |

time, accompaniments are more complex, involving multiple instruments and rarely pausing, leading to discrepancies in timing and pitch between the vocal and accompaniment notes. However, VocalBand still outperforms StyleSinger and achieves satisfactory results. This demonstrates that MultiBand can leverage the user's preferred GT accompaniment for vocal pairing, with VocalBand exhibiting excellent rhythm and melody control by decoupling content information.

# I ABLATION STUDY

## I.1 EXPERIMENTS ON TEXT ENCODER

Table 14: Results of ablation study on different text encoders.

| Methods | FAD ↓ | KLD ↓ | CLAP ↑ | OVL ↑ | REL ↑ |
|---|---|---|---|---|---|
| MultiBand (T5) | **3.03** | **1.26** | **0.55** | **87.66±1.34** | **87.95±0.79** |
| MultiBand (CLAP) | 3.31 | 1.34 | 0.41 | 85.36±1.57 | 86.03±1.39 |
| MultiBand (BERT) | 3.12 | 1.29 | 0.49 | 87.02±0.84 | 87.21±0.83 |

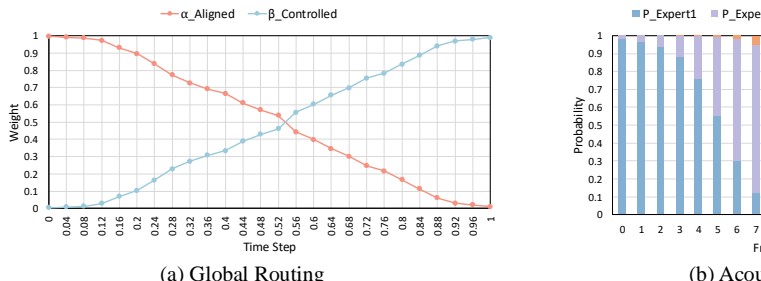

(a) Global Routing                    (b) Acousitc Routing

Figure 11: The statistics of global routing and acoustic routing in Band-MOE.

For the text encoder, following previous work (Zhiqing et al., 2024), we test FLAN-T5-large (Chung et al., 2024), BERT-large (Devlin et al., 2018), and the text encoder of CLAP (Elizalde et al., 2023). Table 14 shows that we test MultiBand without inputting lyrics or music scores. It can be seen that T5 outperforms the other two text encoders in both quality and relevance, but has only a slight advantage over BERT, which is likely due to its larger parameter count and multi-task capability.

### I.2 EXPERIMENTS ON MOE

To demonstrate the effectiveness of our MOE, we conducted experiments on the final routing behavior. As shown in Figure 11 (a), we can observe that our global routing behaves as expected. As the noise level decreases, the weighting of outputs from Aligned MOE and Controlled MOE changes accordingly: 1) At early time steps (near 0), where the hidden representation $h$ is highly noisy, the network prioritizes matching with the vocal for coherent reconstruction, thus the weight of the Aligned MOE is higher. 2) At later time steps (near 1), where $h$ has been largely reconstructed, the network focuses more on refining stylistic details, relying heavily on text prompts, thus the weight of the Controlled MOE is higher.

As shown in Figure 11 (b), the Acoustic MOE also behaves as expected by assigning different experts to different channels. We encode the mel-spectrogram into 20 dimensions through the Aligned Accomp Encoder, resulting in 20 channels and selecting experts for each channel. We perform a statistical analysis of the softmax output probabilities before expert selection. 1) Expert 1 focuses on channels 0 to 7, which include instruments that provide foundational rhythm and depth, such as bass guitars, kick drums, low-frequency percussion, and the lower registers of piano and organ. 2) Expert 2 specializes in channels 4 to 12, capturing the richness of rhythm guitars, mid-range piano notes, and various percussion instruments that contribute to the fullness and body of the music. 3) Expert 3 targets channels 9 to 16, encompassing lead guitars, higher piano octaves, string instruments, and brass instruments. This allows the model to capture melodic elements and intricate harmonics that enhance the expressiveness of the accompaniment. 4) Expert 4 is assigned to channels 14 to 19, focusing on cymbals, hi-hats, flutes, and high-frequency string overtones that contribute to the brightness and airiness of the music.

## J LIMITATIONS AND FUTURE WORK

In this section, we discuss two main limitations of MultiBand and provide potential strategies to address them in future work:

- **Model Complexity.** To achieve comprehensive controllability and high-quality multi-task song generation based on various prompts, MultiBand utilizes four sub-models to generate different components of a song, relying on multiple infrastructures like the flow-based transformer and VAE. This results in cumbersome training and inference procedures. Future work will explore the possibility of using a single model to achieve the same multi-task generation capabilities and controllability.

- **Language Diversity.** Our sampled dataset only includes songs in Chinese and English, lacking diversity. In the future, we will attempt to build a larger and more comprehensive dataset to enable a wider range of application scenarios.

