# OpenReview forum: "MultiBand: Multi-Task Song Generation with Personalized Prompt-Based Control"
_ICLR.cc/2025/Conference — ICLR 2025 Conference Withdrawn Submission_

### Official Review · Reviewer_iytc · 2024-11-01

**Soundness:** 3
**Presentation:** 1
**Contribution:** 2
**Rating:** 3
**Confidence:** 3

**Summary:**

MultiBand is a system that generates pop song phrases directly in the audio domain. The system is divided into two functional blocks: composition and synthesis. The composition block consists of MelodyBand and LyricBand, which "write" music melodies with lyrics. The synthesis block consists of VocalBand and AccompBand, which synthesize singing voices and generate accompaniment, respectively. The main technical focus is on the synthesis block. Specifically, VocalBand leverages style disentanglement to enhance controllability and audio quality; AccompBand utilizes contrastive learning to maintain vocal-accompaniment alignment. Additionally, each module accepts control options from text prompts and audio examples, facilitating personalized generation. Experimental results demonstrate the system’s versatility across various generation and synthesis tasks, with ablation studies validating the effectiveness of the VocalBand and AccompBand designs.

**Strengths:**

* This is a well-engineered system, uniting many functional modules to form a complete pipeline: from melody & lyrics composition, to vocal synthesis and accompaniment generation.

* Leveraging disentanglement and classifier-free guidance, this system (at different module levels) supports flexible control from text prompts and audio examples.

**Weaknesses:**

The reviewer identifies one weakness in the quality of the demo music, specified as follows:
   * Despite the emphasis on "alignment" in Section 3.3, several listening demos seem to exhibit a misalignment, particularly with downbeats and harmonic structure. For example, in the 1st demo piece in Vocal-To-Song Generation, the strong beat in the Generated Accompaniment has an offset to the GT Vocal's downbeat. In the later part of this accompaniment, the chords also sound a bit discordant. Similar cases happen in other generation examples especially *when vocal is condition*. The reviewer also notes that these demos are typically inferred on a short vocal segment (5-10 seconds), where it is indeed more challenging for a model to locate downbeats. Maybe the authors could try inferring on longer samples and see if it benefits, or training and testing using complete n-bar segments (this would require processing dataset using a downbeat detection model, as done in [1]).

[1] K. Chen, et al. Musicldm: Enhancing novelty in text-to-music generation using beat-synchronous mixup strategies, in ICASSP 2024

This paper could also benefit from improved clarity. Details are as follows:
   * This work is an engineering system with complex and interlocked design choices. In the current writing, readers cannot find a structured flow from the technical details, while the intuitive insights are somewhat obscured. It might be better to be more general on "how" the modules function, and more specific on "why" each design choice is necessary or beneficial.
   * Several concepts mentioned in this paper are not well-defined, such as "alignment," "style transfer," and "high-fidelity." Specifically, "alignment" could include "metrical (i.e., beat/downbeat) alignment" as the reviewer has identified, as well as style coherency, timbre matchness, etc. A finer definition of the "alignment" here could help readers better comprehend the function of the Alignment Encoder. A formal evaluation on "alignment" (by either objective or subjective metrics) can further solidify the argument. For "style transfer" and "high-fidelity", the reviewer has raised corresponding questions in the Questions section of this review.

**Questions:**

* On "style transfer:" Based on the reviewer's understanding, "singing style transfer" typically refers to singing A's song using B's voice style, and the "voice style" here could refer to vocal timbre and/or singing techniques [1]. In this paper, the "Singing Style Transfer" task uses the original singer's voice feature, which is more like a reconstruction task. If the authors wish to stick to the term "style transfer," then this term should be defined more formally, and it is suggested to use the voice feature from varied singers. (For the "Music Style Transfer" task, similarly, the accompaniment prompt from varied sources would be expected).

[1] S. Dai, et al. SingStyle111: A Multilingual Singing Dataset With Style Transfer, in ISMIR 2023.

* Lines 072 describe MusicGen as having "lower fidelity" compared to the proposed system. Fidelity is often quantified through objective metrics, such as SNR (Signal-to-Noise Ratio) and SAR (Signal-to-Artifact Ratio), but here, the term is used less formally, more like a promotional language. (Note that MusicGen is capable of generating stereo audio.) The reviewer would thus discourage using "low-" or "high-fidelity" this way without a more formal definition and evaluation. A similar promotional language to prevent is "*perfectly* aligned with vocals" in the abstract.

* Section 4.1 describes dataset collection from multiple sources. Have the authors checked if there are any overlapped music pieces across different datasaets? In the 500 samples with "unseen" singers for testing (Line 336), how does it make sure that the singers are unseen?

* Line 260: seemingly a typo with "audio-video"

---

### Official Review · Reviewer_47m5 · 2024-11-01

**Soundness:** 3
**Presentation:** 2
**Contribution:** 3
**Rating:** 6
**Confidence:** 3

**Summary:**

This paper introduces a system called MultiBand for vocal and accompaniment audio generation. The model consists of four sub-models and, therefore, breaks down the generation into four parts involving the generation of vocal, accompaniment, lyrics, and melody. The four modules of the generation process are interpretable and can be controlled by text or audio prompts. Experiments show a better generation quality and control effectiveness compared to the baseline methods.

**Strengths:**

The proposed song generation framework is straightforward, offering a structure that is likely easier to control compared to end-to-end systems. Despite the complexity involved, the authors have successfully managed to build and integrate this comprehensive system. The experiments are quite comprehensive. The quality of the demo is good (i.e., better than existing).

**Weaknesses:**

1. While the system is complex and the writing is generally clear, the methodology is difficult to follow. Figures 2 and 3 are way too small and challenging to read, and the high-level design, architecture choice, and implementation details are blended together. I hope greater clarity could be achieved by: (1) providing a more abstract diagram illustrating the diagram of the four modules, and (2) organizing the methodology section into clearer subsections for each module. Additionally, a table summarizing the variables, inputs, outputs, training data description, and "whether fine-tuned" of each module would enhance comprehensibility. In the experiment section, it's important to mention explicitly what are subjective metrics and describe the subjective test setting more in the main paper (not in the appendices).
2. The paper contains "style transfer," but the concept of "style" in this study lacks a clear definition. It’s unclear what specific style elements are being transferred, and it feels more like just a prompt with unpredictable outcomes.
3. The term "personalized" appears in the title and throughout the paper, but its meaning is unclear. The paper does not seem to include an algorithm that adapts to user preferences, making the personalization claim feel unsubstantiated.
4. The generated vocal tracks are often off-tune and occasionally lack coherence with the accompaniment. This may be due to the absence of harmony constraints in the generation process, which is essential for most pop music and could help align vocals and accompaniment more harmoniously.
5. The related work about music generation is far from comprehensive. Consider narrowing the section title. It's also helpful to provide a background on the flow-based approach.

I'll consider raising the point if the weaknesses are improved (especially the first one).

**Questions:**

Questions are mostly mentioned in the weaknesses. One additional question: why are generated singing voices usually out-of-tune?

---

### Official Review · Reviewer_NrP3 · 2024-11-04

**Soundness:** 2
**Presentation:** 2
**Contribution:** 2
**Rating:** 3
**Confidence:** 5

**Summary:**

This paper presents a comprehensive song generation pipeline named MultiBand, which is composed of three main components: (1) **LyricBand and MelodyBand** generate lyrics and melody, utilizing a fine-tuned Qwen-7B model and a non-autoregressive transformer MIDI generator, respectively; (2) **VocalBand**: with the lyrics and melody in place, VocalBand employs singing voice synthesis (SVS) techniques to generate the singing voice, optionally conditioned on a vocal prompt or text description, relying primarily on a flow-based model; (3) **AccompBand**: following the generation of the singing voice, AccompBand produces accompaniment, with the option to condition on an accompaniment prompt. The authors trained the pipeline on a dataset of over 1,000 hours, compiled from in-the-wild sources, LP-MusicCaps-MSD, and others.

**Strengths:**

1. The paper establishes a fairly comprehensive song generation pipeline, covering multiple tasks such as lyric generation, symbolic music generation, singing voice synthesis, and sing-to-song generation. It touches on the main stages of AI music creation relevant to real-world applications.

2. The study addresses several important issues in music generation, such as using text descriptions as conditions to enhance controllability and incorporating various audio prompts as imitation references, which increases accessibility for amateur users.

3. The experimental setup is fairly complete, and the experiments are thorough.

**Weaknesses:**

The problem and motivation of this paper are well-chosen (though these motivations and ideas are not proposed firstly in this study). The main weaknesses are in the methodological design and the final model performance.

1. In terms of method and results, the primary innovation of this work seems to lie in the design of VocalBand (although AccompBand introduces some minor innovations, such as using a mixture of experts (MoE), the impact appears minimal). VocalBand essentially adds "zero-shot" (vocal prompt) and "instruction" (style prompt) capabilities to a standard flow-matching singing voice synthesis model. However, these design elements are already well-known and commonly used in the TTS field.

2. For the accompaniment generation task, based on the demo page results, I find that MultiBand's performance does not align with the claims made in Table 4. Issues such as the misalignment of accompaniment and singing voice downbeats appear unresolved. Moreover, the authors frequently highlight "high-quality generation" throughout the paper. While they used 48kHz mel-spectrograms, the generated results still fall short of high-quality sound (arguably even below the acoustic quality of 24kHz TTS models). Based on my subjective impression, MultiBand's output quality seems inferior to that of MusicGen [1], despite the results in Table 4 suggesting it outperforms MusicGen.

> [1] https://audiocraft.metademolab.com/musicgen.html

**Questions:**

1. In the samples on the demo page, it appears that the generated vocal inputs include musical notes (melody). Was this part not displayed? For the singing style transfer part, there is a significant amount of in-the-wild singing voice data (presumably source-separated from music recordings?). How were the musical notes for this data obtained?

2. For VocalBand, the introduction of a style prompt input is a relative novel aspect. However, the description in the paper is too brief. How was the ground truth for this style derived?

---

### Official Review · Reviewer_Ts6f · 2024-11-04

**Soundness:** 3
**Presentation:** 3
**Contribution:** 3
**Rating:** 5
**Confidence:** 3

**Summary:**

The paper introduces MultiBand, a novel multi-task model for generating complete songs that integrate vocals and accompaniments. It uses personalized, prompt-based control, allowing for extensive customization based on text and audio prompts. The main components of MultiBand include:

VocalBand: A model that uses flow-matching techniques to generate singing styles, pitches, and mel-spectrograms, enabling high-quality vocal synthesis with fine-grained control.
AccompBand: A flow-based transformer that creates aligned, high-fidelity accompaniments with mechanisms like an Aligned Vocal Encoder and Band-MOE for expert selection.
LyricBand and MelodyBand: Models that generate lyrics and melodies, contributing to a fully integrated song generation system.

The system is designed to overcome challenges like aligning vocals and accompaniments, maintaining musical style coherence, and supporting a wide range of personalized music generation tasks.

**Strengths:**

Extensive Personalization with flexible controls: MultiBand allows for detailed control over various aspects of song composition, such as lyrics, melody, singing styles, and music genres, using a range of customizable prompts. This level of personalization is unmatched in previous models.

A great deal of multimodal LLM engineering: I appreciate the amount of careful engineering.

**Weaknesses:**

As a music generation paper, the most important aspect is the quality of the music generated. Despite all the flexible controls, the primary function—vocal and accompaniment generation—is unfortunately not good enough after listening to the demo. Specifically, the accompaniment seems to follow different chords and barely stays in harmony with the vocal melody.

Note that controlling audio-based LMs with melody and chords (even drums and piano controls) is more or less a solved problem. (Please refer to "Content-based Controls For Music Large Language Modeling" and its related studies) Maybe incorporating more methods could yield better results.

**Questions:**

Have you conducted a study on how often the model copies the training set, especially the accompaniment part? A similar experiment can be found in MusicLM.

---

### Note · Authors · 2024-11-19

I have read and agree with the venue's withdrawal policy on behalf of myself and my co-authors.